# Severing Spurious Correlations with Data Pruning

**Varun Mulchandani & Jung-Eun Kim**[*]
Department of Computer Science
North Carolina State University
{vmmulcha,jung-eun.kim}@ncsu.edu

## Abstract

Deep neural networks have been shown to learn and rely on spurious correlations present in the data that they are trained on. Reliance on such correlations can cause these networks to malfunction when deployed in the real world, where these correlations may no longer hold. To overcome the learning of and reliance on such correlations, recent studies propose approaches that yield promising results. These works, however, study settings where the strength of the spurious signal is significantly greater than that of the core, invariant signal, making it easier to detect the presence of spurious features in individual training samples and allow for further processing. In this paper, we identify new settings where the strength of the spurious signal is relatively weaker, making it difficult to detect any spurious information while continuing to have catastrophic consequences. We also discover that spurious correlations are learned primarily due to only a handful of all the samples containing the spurious feature and develop a novel data pruning technique that identifies and prunes small subsets of the training data that contain these samples. Our proposed technique does not require inferred domain knowledge, information regarding the sample-wise presence or nature of spurious information, or human intervention. Finally, we show that such data pruning attains state-of-the-art performance on previously studied settings where spurious information is identifiable.[1]

## 1 Introduction

Deep neural networks have shown promising results on a variety of benchmarks and applications. However, their reliability and by extension, ability to solve increasingly challenging problems remains questionable as these networks have been shown to exhibit several failure modes in practice. Of these, the inability to adapt to distributional shifts due to the reliance on spurious correlations is of strong concern as it makes them unreliable for deployment in the real world. Spurious Correlations are correlations that a network learns between simple, weakly predictive spurious features present in a fraction of the training data and the class label. These correlations are problematic as a network may prefer them over strongly predictive invariant correlations when making a prediction. Thus, in the event of a distribution shift where spurious features either no longer exist or become correlated with a different task, which is a common phenomenon in the real world, these networks begin to malfunction (Arjovsky et al., 2019; Sagawa et al., 2020a;b; Nagarajan et al., 2021; Geirhos et al., 2020).

To promote the learning of invariant features, several recent works have proposed techniques that yield promising results. These works heavily rely on sample-wise environment labels, where each class is spuriously correlated with a spurious feature from one environment. Such labels are also referred to as group labels. For instance, Sagawa et al. (2020a) aim to prioritize samples from environments that have high training risk to promote invariant feature learning, with the assumption that such

---

[*]Corresponding author.
[1]Code is available at: https://github.com/JEKimLab/ICLR2025_SpuriousDataPruning

labels are available. Kirichenko et al. (2023) work with a stronger assumption, where they assume that overrepresented environments contribute to the learning of spurious correlations and simply re-train the last layer of a spuriously biased network on a dataset where all environments are equally represented. Such a solution makes strong assumptions regarding which samples contain spurious features associated with each class. Moayeri et al. (2023) follow a similar technique of utilizing human supervision to determine the strength of spurious signals in each training sample and fine-tune a biased network on samples where spurious signals are weaker. Attaining sample-wise environment labels and information regarding sample-wise presence of spurious features during training, however, is unrealistic. To overcome this problem, Liu et al. (2021); Zhang et al. (2022); Pezeshki et al. (2024) aim to infer the sample-wise presence of spurious features based on a biased network's outputs and they follow this with further processing such as up-weighting of samples without spurious features or aligning representations of samples with and without spurious features, typically at the cost of overall testing accuracy.

All past works that study spurious correlations and aim to promote invariant feature learning perform experiments on settings where the strength of the spurious signal is significantly higher than the core, invariant signal. This makes it easy to identify the presence of spurious features in each sample within the training set, which is, however, not always the case. In this paper, we aim to tackle novel settings where the strength of the spurious signal is relatively weaker, making it impossible to identify such information while rendering existing techniques ineffective. We then show that spurious correlations are learned primarily due to a few key samples in the training set and present a novel data pruning technique to identify and prune a small subset of the training data that contains these samples. Finally, we show that such data pruning attains state-of-the-art performance on previously studied settings where information regarding the sample-wise presence of spurious features is easily identifiable. We summarize our contributions below:

**Contributions**

- We identify settings where it is impossible/difficult to identify sample-wise presence of spurious features. This renders past approaches that mitigate spurious correlations as ineffective.

- We discover that spurious correlations are learned primarily due to a handful of all the samples containing spurious features, through extensive empirical investigation. Based on this insight, we propose a simple and novel data pruning technique that identifies and prunes a small subset of the data that contains these samples.

- We show that such data pruning attains state-of-the-art results even in previously studied settings where information regarding the sample-wise presence of spurious features is identifiable.

## 2 BACKGROUND AND RELATED WORK

Consistent with past literature, we study the supervised classification setting where $S = \{(x_i, y_i)\}_{i=1}^N$ denotes the training dataset of size $N$ and network is trained to learn a mapping between $x_i$ (input) and $y_i$ (class label) using empirical risk minimization (Vapnik, 1998). Every training sample $s \in S$ contains a core feature $(c_i)$ that represents its class $(y_i)$. A fraction of all samples within a class contain the spurious feature $(a_i)$ associated with that class. Core (or invariant) features represent the class label $y_i$ and are semantically relevant to the task. They are also fully predictive of the task, as they are present in all samples. Spurious features do not represent the class labels and are semantically irrelevant to the task.

**Spurious Correlations.** The correlations a network learns between spurious features and class labels. Such correlations are undesirable as they can disappear during testing or become associated with a different task, causing these networks to malfunction (Arjovsky et al., 2019; Nagarajan et al., 2021; Shah et al., 2020; Tsipras et al., 2019; Sagawa et al., 2020b;a; Kirichenko et al., 2023; Jain et al., 2023; Ye et al., 2022). To make a network robust against spurious correlations, Sagawa et al. (2020a) suggest using information regarding sample-wise environment labels during training. Since such information is often unavailable, more recent approaches attempt to infer group or spurious information with the help of an ERM-trained model, followed by subsequent sample up-weighting or class-wise representational alignment (Sohoni et al., 2020; Liu et al., 2021; Zhang et al., 2022;

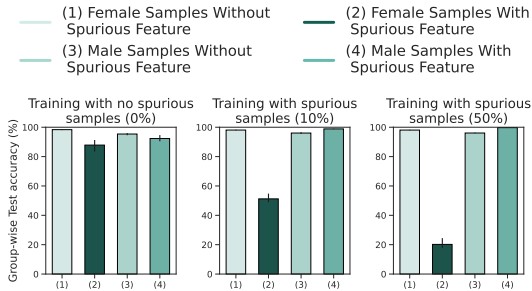
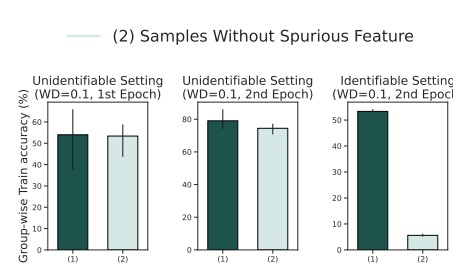

(a) Training with just a *few* Male samples with spurious features causes heavy reliance on spurious features, indicated by low test accuracy on Female samples with Spurious Features.

(b) Training with greater regularization may not help identify samples with the spurious feature.

Figure 1: Spurious information is often unattainable.

Ahmed et al., 2021; Creager et al., 2021; Pezeshki et al., 2024). Recent work has also shown that a model learns both general and spurious features and just re-training the last layer with balanced data where strong spurious correlations do not hold can improve robustness (Kirichenko et al., 2023). Moayeri et al. (2023) perform a similar re-training, where they fine-tune the final layers of trained (biased) models on those samples with minimal spuriosity, where spuriosity is determined with human supervision. Idrissi et al. (2022) train a network on a balanced train set where all environments are equally represented, thereby promoting invariant feature learning.

**Simplicity Bias and Spurious Correlations.** Recent work has shown that deep neural networks have a greater affinity towards simpler features than more complicated ones (Shah et al., 2020; Morwani et al., 2023; Tiwari & Shenoy, 2023). Shah et al. (2020) show that in the presence of two fully predictive features, a network would choose to fully ignore the more complicated features in favor of the simpler set of features. In settings where the simpler feature is not fully predictive (Spurious), the network still relies strongly on these features (Kirichenko et al., 2023), leading to the learning of and reliance on spurious correlations.

**Feature Difficulty.** Consistent with deep learning literature (specifically, those works concerned with spurious correlations), difficulty of learning a feature is determined by the following three factors: (1) Proportion (or Frequency) of training samples containing the spurious feature (Sagawa et al., 2020b; Shah et al., 2020; Kirichenko et al., 2023), (2) Area Occupied and Position (if it is centered or not) in the training sample (Moayeri et al., 2022) and (3) The amount of noise in its signal (Sagawa et al., 2020b; Ye et al., 2022). A feature which is present in a large portion of all training samples, occupies a lot of area, is centered, and has little to no variance, is easy to learn. On the other hand, a feature which is present in a small portion of all training samples, occupies little area, is not centered, and has a lot of noise/variance, is hard to learn.

Our work is concerned with two different widely studied areas in deep learning: out-of-distribution generalization and data pruning.

**Data Pruning.** A large part of deep learning's recent success has been attributed to large datasets that models are trained on. However, with increasing datasets sizes, computational costs have risen significantly. This raises an important question: Are all samples in a dataset equally important for attaining good testing accuracy? In other words, is it possible to reduce samples in the training data without impacting generalizability? Recent work has shown that it is possible to remove a large fraction of a dataset without sacrificing test accuracy. Most approaches create sample-wise metrics that measure how important it is to maintain a given sample in the training set and remove the least important samples by their definition (Paul et al., 2021; Toneva et al., 2019; Feldman & Zhang, 2020; Sorscher et al., 2022; He et al., 2023).

While existing works aim to prune samples to reduce computational costs, our work focuses on pruning samples to mitigate the learning of spurious correlations during training. This direction, to the best of our knowledge, is *novel*.

## 3 Spurious Information is Often Unattainable

Suppose one wants to build a gender classifier using the CelebA dataset (Liu et al., 2015). A small fraction of the training samples in the Male class contain eyeglasses (10%), which takes the form of the spurious feature in our setting. None of the samples in the Female class contain eyeglasses in the training set. Thus, the Male class is spuriously correlated with eyeglasses. Note that in our setting, only a small minority of all Male samples contain the spurious feature, making the strength of the spurious signal relatively weaker. This is different from settings studied in literature, where the spurious feature is present in a majority of the samples (often 95-97% of all samples of that class) and the strength of the spurious signal is significantly higher than the strength of the core, invariant signal. We observe that under normal training conditions without the existence of spurious samples, test accuracy of Female samples with the spurious feature is high as shown in Fig. 1a (Left). Introduction of just a few Male samples with the spurious feature is capable of reducing the test accuracy of Female samples with eyeglasses significantly, as shown in Fig. 1a (Middle and Right).

In settings where the strength of the spurious signal is significantly greater than the strength of the core, invariant signal, it becomes easy to differentiate between groups of samples containing spurious features and those that do not (Liu et al., 2021; Zhang et al., 2022; Yang et al., 2024). This is because the network relies very strongly on the spurious features during training, making it easy to differentiate between such groups based on representational differences or on a network's ease of learning these samples. For instance, Liu et al. (2021); Zhang et al. (2022) train a network with high regularization (greater weight decay) such that a network correctly classifies samples with easy spurious features early during training and misclassifies those without, as shown in Fig. 1b (Right) - The identifiable setting shown is the CelebA setting studied in Liu et al. (2021). In our setting, however, the spurious feature is not present in a majority of the training samples and so the strength of the spurious signal is relatively weaker. This makes it impossible to identify which samples contain spurious features using past approaches, even when 50% of all male samples contain eyeglasses - see Fig. 1b (Left and Middle). Thus, we identify the primary failure mode of existing approaches as follows:

- *Attaining information regarding the presence of spurious features becomes difficult when the strength of the spurious signal is not exceptionally greater than the strength of the core, invariant signal in the training set.*

Based on this observation, we propose the following problem statement:

- **Problem Statement:** *How does one overcome spurious correlations in settings where attaining spurious information is difficult or impossible?*

## 4 Experimental Design

In this paper, we study both settings: those where information regarding spurious information is identifiable and those where such information is unidentifiable. To support our claims, we study both Vision and Language Tasks. We also move beyond simple binary classification tasks that have been the focus of all recent works that study spurious correlations. Below, we detail the experimental design for this paper. Please note that we provide additional details in the Appendix.

Testbed

- **CIFAR-10S.** We create a testbed based on the CIFAR-10 dataset (Krizhevsky, 2009) where we synthetically introduce spurious features in a fraction of one class' ($c1$) training samples. We introduce the same spurious feature in all samples of a different class ($c2$) during testing. The degree of spurious feature reliance is estimated by measuring the number of samples of ($c2$) that are misclassified as ($c1$) during testing. Since spurious features are synthetically introduced, this setting allows us to:
  1. Vary the strength of the spurious feature relative to the core, invariant feature.
  2. Accurately compute the difficulty of learning the core, invariant feature of individual training samples before the introduction of spurious features.

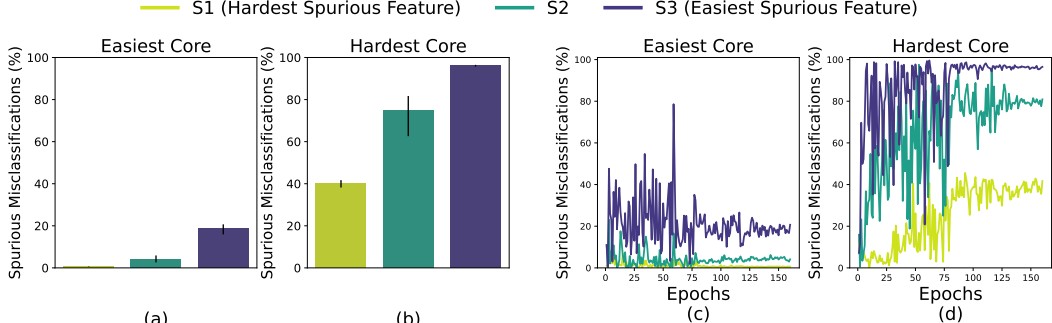

Figure 2: Introducing spurious features in 100 samples with the easiest core features (`Easiest Core`) causes little to no reliance on spurious features, indicated by low Spurious Misclassifications. Introducing the *same* spurious features in 100 samples with the hardest core features (`Hardest Core`) causes heavy reliance on spurious features, indicated by high Spurious Misclassifications.

Note that in this setting, the spurious feature takes the form of a line running through the center of the images of class $c1$ and we vary the strength of the spurious signal by increasing the region the spurious feature covers or by increasing the proportion of samples that contain the spurious feature. We note that in each experimental setting studied, we only vary one of the two factors.

UNIDENTIFIABLE BENCHMARKS

- **CelebA.** (Liu et al., 2015) We utilize the same experimental setting studied in Sec. 3, where we build a gender classifier in which the `Male` class is spuriously correlated with `Eyeglasses`. The degree of spurious feature reliance is measured based on the testing accuracy of `Female` samples with `Eyeglasses`.

- **Hard ImageNet.** (Moayeri et al., 2022) The Hard ImageNet dataset is a 15 class classification task where all classes have a spurious feature associated with them and certain classes share similar spurious features. For instance, both `Ski` and `Dog Sled` classes contain similar spurious features (`Snow`, `People`, `Trees`, and `Hills`). In our experiments, we measure the degree of spurious feature reliance by measuring the numbers of `Dog Sled` samples that are misclassified as `Ski` during testing. Note that in our experimental setting, no `Skis` are misclassified as `Dog Sled`.

STANDARD BENCHMARKS (IDENTIFIABLE)

- **Waterbirds.** (Sagawa et al., 2020a) The Waterbirds task is a binary image classification task where the goal is to classify an image of a bird as `landbird` or `waterbird`. We use the same setting studied in previous works (Sagawa et al., 2020a; Kirichenko et al., 2023), where the class `landbird` is spuriously correlated with `Land` backgrounds while the class `waterbird` is spuriously correlated with `Water` backgrounds.

- **MultiNLI.** (Williams et al., 2018) The MultiNLI task is a classification task with three classes where the goal is to classify the second sentence in a pair of sentences as entailed by, neutral with, or contradicts. Consistent with the setting in Sagawa et al. (2020a), a large fraction of the `contradicts` class contains negation words while the other two only contain a few samples with negation words, making the `contradicts` class spuriously correlated with negation words and the other two with the lack thereof.

EVALUATION.

Current practice in deep learning utilizes Worst-Group Accuracy (WGA) to assess the degree of spurious feature reliance in binary classification tasks. WGA computes the accuracy of test samples that contain the spurious feature associated with the other class during training. While suitable for simple binary classification tasks, WGA becomes insufficient to asses the reliance on spurious

features in settings with multiple classes. This is because WGA cannot differentiate between loss in test accuracy due to spurious correlations, or due to lack of learnability of invariant correlations stemming from limited capacity or insufficient training data. In such settings, we measure the degree of spurious feature reliance through *Spurious Misclassifications*, *i.e.*, the percentage of samples of one class ($c2$) containing the spurious feature of another class ($c1$) that are misclassified as ($c1$) during testing. Lower Worst Group Accuracy indicates heavy reliance on spurious correlations while high Worst Group Accuracy indicates little to no reliance on spurious correlations. A high number of Spurious Misclassifications indicates heavy reliance on spurious correlations while a low number Spurious Misclassifications indicate little to no reliance on spurious correlations.

## 5 SPURIOUS CORRELATIONS ARE LEARNED FROM A FEW KEY SAMPLES

In this section, we aim to form a deeper understanding of the relationship between spurious correlations and standard training paradigms. Deep neural network training relies on the Empirical Risk Minimization (Vapnik, 1998) principle, where during training, the model attempts to minimize risk or loss on the training data. From recent works, we understand that spurious correlations are learned because the network uses these weakly predictive but simple spurious features to further minimize training risk (Arjovsky et al., 2019; Sagawa et al., 2020a; Geirhos et al., 2020; Kirichenko et al., 2023). In this section, we attempt to understand if this simple statement encapsulates the entire relationship between deep neural network training and spurious correlations.

We understand that spurious correlations are learned because standard training paradigms are prone to accepting any features that help minimize risk or loss during training. However, different samples have different contributions to overall training loss due to differences in difficulty of learning the features used to make correct classification. Does this imply that samples containing harder core/invariant features contribute more to the learning of spurious correlations? Or do all samples contribute equally to the learning of spurious correlations, as a network might simply prefer a simpler feature over a more complex one?

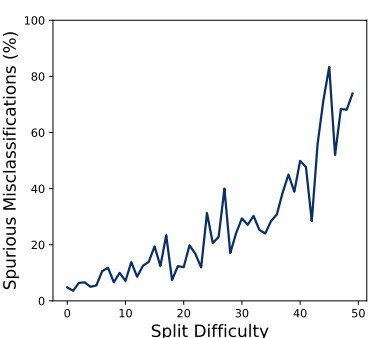

Figure 3: Spurious feature reliance exhibits super-linear growth with increasing sample difficulty.

To answer this question, we study a variant of the CIFAR-10S setting described in Sec. 4. In this experiment, we first train a ResNet20 on standard CIFAR-10. Early in training, we compute training error for each sample to estimate the difficulty of learning these samples, as is done in Paul et al. (2021). Such sample-wise training error is computed as $||p(w, x) - y||_2$, where $p(w, x)$ is the probability distribution given by the network for sample $x$, $w$ denotes the network parameters early in training, and $y$ is the one hot encoding of the ground truth value. We specify the epoch at which these values are computed in the Appendix. Trivially, the greater the error, the more difficult that sample is to learn. Since the standard CIFAR-10 does not contain significant spurious cues that can impact difficulty of learning, we estimate the difficulty of learning the core/invariant feature per sample as the difficulty of learning that sample. Next, we train a ResNet20 on two settings: One where we introduce the spurious feature in samples with the lowest training error (samples with *easy invariant* features) and another where we introduce spurious features in samples with the highest training error (samples with *hard invariant* features). We term these settings as `Easiest Core` and `Hardest Core`, respectively. In each setting, we vary the strength of the spurious signal by increasing the amount of area it takes up in each image. Spurious feature $S1$ takes up the least amount of area and is the most difficult to learn while spurious feature $S3$ takes up the most amount of area and is the easiest spurious feature to learn. In other words, a network will have less reliance on spurious feature $S1$ as it is difficult to learn and more reliance on $S3$ as it is easier to learn. $S2$ is in between them. Note that in both experimental settings, we introduce spurious features in only 100 samples (2% of class $c1$, 0.2% of the training set), and thus, we maintain the proportion of samples containing the spurious features in all settings.

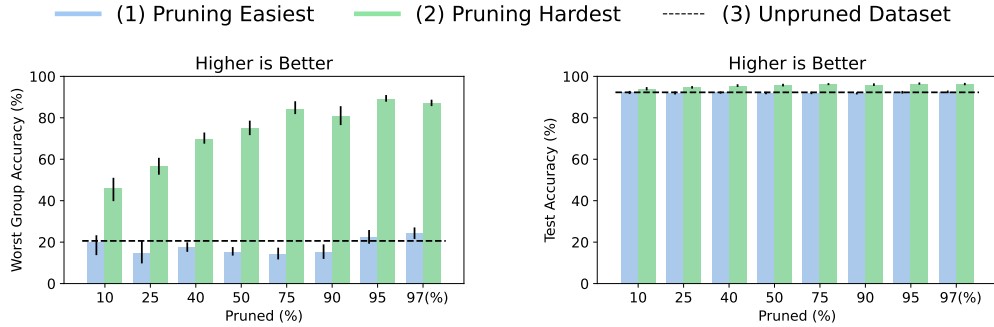

Figure 4: Excluding only a handful of training samples with spurious features and hard core features mitigates spurious correlations in the CelebA setting. This is indicated by high Worst Group Accuracies. Excluding up to 97% of all training samples with spurious features and easy core features shows no improvements in worst group accuracy.

**Samples with simple core features do not contribute to spurious correlations.** In the setting where we introduce spurious features in samples with low training loss (`Easiest Core`), we find that Spurious Misclassifications tend to zero (Fig. 2(a)). This is true even if the strength of the spurious signal is increased. More interestingly, the network learns the spurious features present in these samples in the first couple of epochs of training but overcomes them as training converges. This is evidenced by high Spurious Misclassifications in the first half of training followed by low Spurious Misclassifications in the second half of training, as shown in Fig. 2(c). In other words, the network learns to ignore weakly predictive but simpler spurious features and prefers strongly predictive but complex core features in those samples that contain relatively easier invariant features.

**Samples with hard core features are primary contributors to spurious correlations.** In the setting where we introduce spurious features in samples with high training loss (`Hardest Core`), we find that Spurious Misclassifications are significantly higher, as shown in Fig. 2(b). This is also true for spurious features that are harder to learn ($S1$). Unlike what we observe in the previous setting, the network is unable to overcome the spurious features present in these samples, as we observe that Spurious Misclassification either remains the same or increases in the second half of training, as shown in Fig. 2(d). It is important to note that although only 100 of all samples of class $c1$ contain the spurious feature (2% of class $c1$, 0.2% of the training set), almost all samples of class $c2$ ($\approx 100\%$ of class $c2$) are spuriously misclassified during testing (Fig. 2(b)). This is in sharp contrast to the setting where we introduce **the same** spurious features in samples with easy invariant features (`Easiest Core`), where almost **none** of the samples of class $c2$ are spuriously misclassified (Fig. 2(a)).

**Spurious feature reliance exhibits super-linear growth with increasing sample difficulty.** We extend our original experimental setting to introduce the spurious feature in 100 samples ranging from the easiest to the hardest, instead of only considering the Easiest and Hardest 100 samples. Intuitively, as we scale up the difficulty of the 100-sample split to which we inject the spurious features, we observe an increase in spurious misclassifications, as shown in Fig. 3. Interestingly, we observe that spurious feature reliance exhibits super-linear growth as we scale up the difficulty of the core features of the 100 samples, implying that those samples with hard core features contribute significantly more to spurious feature reliance than those with easy core features.

**Excluding a few key samples during training severs spurious correlations.** From the experiments above, we learn that samples with harder core features contribute far more to spurious feature reliance than samples with easy core features. Consider a setting where spurious features are uniformly distributed across samples containing easy and hard core features. In such a setting, what if we simply exclude a handful of the hardest training instances that contain spurious features in them? Does this simple exclusion sever the learned spurious correlations that would be heavily relied on by the network? In Fig. 4, we show that by simply excluding a few samples containing the **spurious** feature with **hard invariant** features in the CelebA setting studied in Sec. 3 (10% of all samples with spurious features in that class, 1% of the total train set), we observe significant improvements in Worst Group Accuracy (WGA: Test accuracy of Female samples with spurious features). On the other hand, excluding up to 97% of the easiest samples containing the spurious feature shows

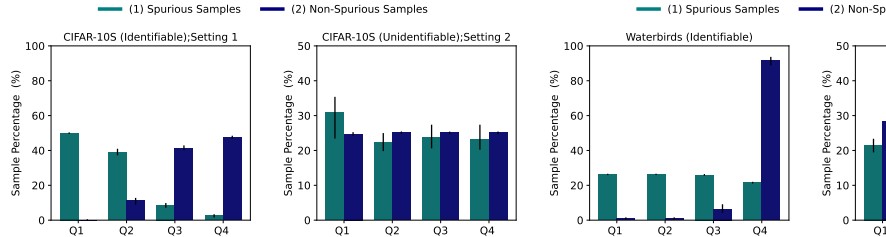

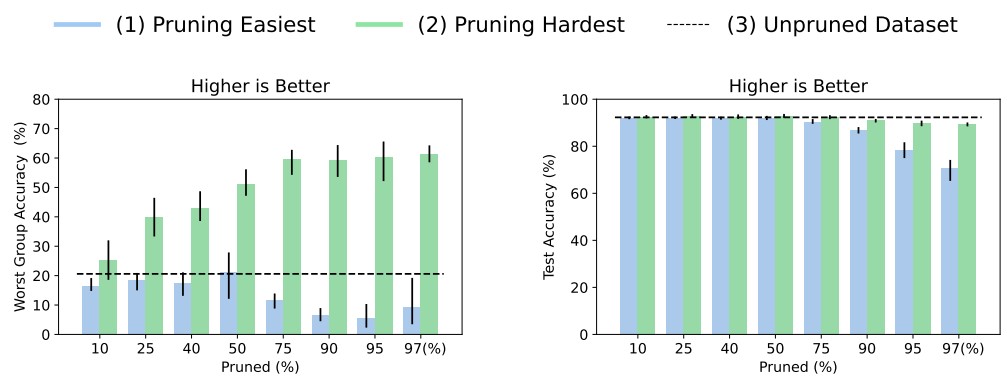

(a) Training distribution variance based on the strength of the spurious feature in CIFAR-10S Testbed. Grouped by Quartiles, sorted by difficulty.

(b) Training distribution variance based on strength of spurious feature in identifiable and unidentifiable settings. Grouped by Quartiles, sorted by difficulty.

Figure 5: Impact of strength of spurious signal on sample difficulty.

Figure 6: Excluding a small fraction of all hardest samples in the dataset mitigates spurious correlations in the CelebA setting. This is indicated by high Worst Group Accuracies.

no improvements in WGA. Within the pool of samples containing spurious features, difficulty of learning the invariant feature is similar to the difficulty of learning the sample. This is because spurious features in our settings exhibit very low variance. For instance, in the MultiNLI setting, the spurious feature comprises the same set of negation words across the training data. Additionally, we note that on pruning these samples, we do not observe significant drops in overall testing accuracies, implying that these samples do not contribute significantly to generalizability either (Fig. 4).

## 6 SEVERING SPURIOUS CORRELATIONS WITH DATA PRUNING

Based on the observations presented above, we develop a novel data pruning method to sever learned spurious correlations by identifying and pruning subsets of the training data that contain the few key samples that contribute to spurious feature reliance. To do so, we first understand how the presence of spurious features impacts the difficulty of learning a sample in the training set and how it impacts the training distribution when observed through the lens of sample difficulty.

### 6.1 THE IMPACT OF SPURIOUS FEATURES ON TRAINING DISTRIBUTION

Consider the CIFAR-10S setting described in Sec. 4. We create two versions of this setting: **Setting 1**: the strength of the spurious signal is significantly greater than the strength of the core, invariant signal; **Setting 2**: the strength of the spurious signal is only marginally greater than the strength of its invariant counterpart. In Setting 2, spurious information is unattainable/unidentifiable. In both settings, we introduce spurious features into samples at random such that both samples with easy and hard core features contain spurious features in them. We observe in Setting 1, almost all of the samples that contain the spurious feature lie in the first half of the data distribution when observed through the lens of sample difficulty. In Setting 2, however, samples containing spurious features are

distributed uniformly (Fig. 5a). Note that in both settings, spurious misclassifications are significant (58.0% and 13.76% in Setting 1 and Setting 2, respectively).

We extend these findings to the benchmarks discussed in Sec. 4. In settings where the strength of the spurious signal is significantly greater than the strength of the invariant feature, (`Identifiable settings`), samples containing the spurious feature occupy a large majority of the first half of the training distribution. On the other hand, in the **unidentifiable** benchmarks, samples containing spurious features are **uniformly distributed** across the training distribution when viewed through the lens of sample difficulty (Fig. 5b). Note that since Hard ImageNet has multiple spurious features, we show that this setting is unidentifiable by showing images with and without these spurious features spread uniformly across the difficulty spectrum in the Appendix.

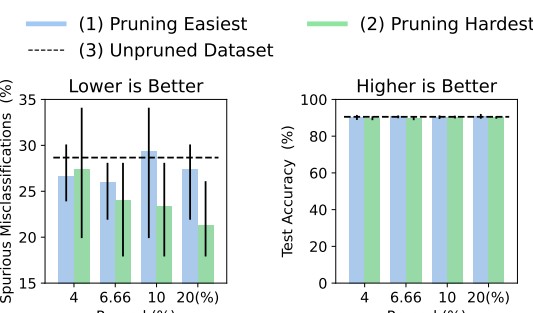

Figure 7: Excluding a small fraction of all hardest samples in the dataset mitigates spurious correlations in the Hard ImageNet setting. This is indicated by low Spurious Misclassifications.

## 6.2 DATA PRUNING IN UNIDENTIFIABLE SETTINGS

From Sec. 6.1, we observe that in settings where the strength of the spurious signal is not significantly greater than the strength of the invariant signal, i.e. in settings where attaining information regarding the sample-wise presence of spurious features is impossible, samples containing spurious features are uniformly distributed across the training distribution when observed through sample difficulty. In other words, the presence of spurious features does not have a significant impact on the training distribution. We also know that samples containing hard core features that also contain the spurious feature are primary contributors to the learning of spurious correlations. Thus, to mitigate spurious correlations without knowing which samples have spurious features in them, one would only have to prune the **hardest** samples in the training set, as this subset of the data would contain samples with **spurious** features that have **hard core** features.

In Figs. 6 and 7, we show that by simply pruning a small subset of the hardest samples in the training set, one can overcome spurious correlations. Note that instead of pruning the hardest samples globally, we prune an equal proportion of the hardest samples per class to account for differences in difficulty per class. *Most importantly, we do not perform any hyperparameter-tuning.* Gulrajani & Lopez-Paz (2021) demonstrate that state-of-the-art techniques fail without hyperparameter-tuning with the help of a validation split that mimics test-time distribution. Our data pruning technique shows promising results without any hyperparameter tuning.

Table 1: We present Worst Group and Mean accuracies. Consistent with literature, Mean accuracy is reported by weighting groups based on their prevalence in the unpruned training dataset. Simple data pruning attains state-of-the-art performance on standard benchmarks (Identifiable Settings). The group labels column represents the availability of group labels in training and validation sets.

| | Waterbirds (%) | | MultiNLI (%) | | Group Labels | |
|---|---|---|---|---|---|---|
| Method | Worst% | Mean% | Worst % | Mean% | Train | Val |
| ERM | 74.81 (0.7) | 98.10 (0.1) | 65.9 (0.3) | 82.8 (0.1) | ✗ | ✗ |
| CnC (Zhang et al., 2022) | 88.5 (0.3) | 90.9 (0.1) | - | - | ✗ | ✓ |
| JTT (Liu et al., 2021) | 86.7 | 93.3 | 72.6 | 78.6 | ✗ | ✓ |
| gDRO (Sagawa et al., 2020a) | 86.0 | 93.2 | **77.7** | 81.4 | ✓ | ✓ |
| DFR[Tr] (Kirichenko et al., 2023) | 90.2 (0.8) | 97.0 (0.3) | 71.5 (0.6) | 82.5 (0.2) | ✓ | ✓ |
| PDE (Deng et al., 2023) | 90.3 (0.3) | 92.4 (0.8) | - | - | ✓ | ✓ |
| **Ours** | **90.93 (0.58)** | 92.48 (0.72) | 75.88 (1.62) | 81.07 (0.25) | ✓ | ✓ |

## 6.3 DATA PRUNING IN IDENTIFIABLE SETTINGS

Unlike in unidentifiable settings, in settings where the strength of the spurious signal is significantly greater than the strength of the invariant signal, samples with spurious features are not uniformly distributed when sorted by sample difficulty. This is because the presence of strong spurious information enables the network to have lower training errors for samples with hard core features and spurious features. In such settings, it is not feasible to simply prune the hardest samples of the training data as this will primarily prune samples that do not contain the spurious features (Figs. 5a & 5b). Liu et al. (2021); Zhang et al. (2022); Yang et al. (2024) show that in settings where the strength of the spurious signal is significantly greater than the strength of the invariant signal, it is possible to identify which samples and groups contain spurious

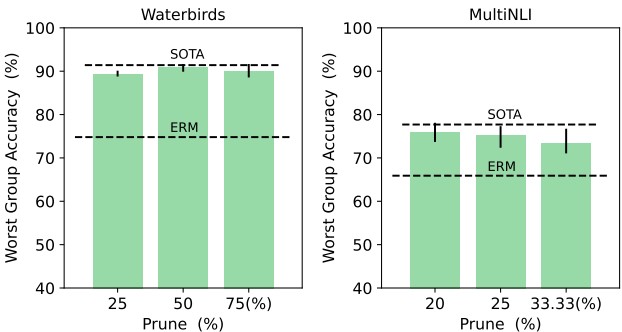

Figure 8: Our data pruning approach is effective across a wide range of sparsities. Sparsities reported before class balancing, if applicable. We report SOTA results for Waterbirds and MultiNLI from Kirichenko et al. (2023) and Sagawa et al. (2020a), respectively. Note that Kirichenko et al. (2023) obtain state-of-the-art results on Waterbirds with the help of a held-out group balanced dataset (DFR$^{\text{Val}}$) and thus, we exclude it from our comparison in Table 1.

features in them and which ones do not. For such identifiable settings, our approach works with group labels as is done in Kirichenko et al. (2023); Deng et al. (2023) and simply prunes those samples containing the hardest core features within groups containing the spurious feature associated with each class. As shown in Table 1, we attain state-of-the-art performances. Note that in settings with heavy class imbalances, such as Waterbirds, we prune such that the total number of samples in both classes is the same. We also compare our method with two popular techniques that make use of inferred sample-wise spurious information during training, `JTT` (Liu et al., 2021) and `CnC` (Zhang et al., 2022). It is interesting to note that such data pruning attains state-of-the-art performances across a wide range of pruning sparsities, as observed in Fig. 8. This suggests, in addition to all previously observed results, that if one wants to ensure that the model they obtain after training is robust to spurious correlations, they only need to remove a few training instances that are hard for the network to understand. This is in contrast to existing methods that mitigate spurious correlations which are more complex and at times, computationally expensive (Sagawa et al., 2020b; Ahmed et al., 2021; Ye et al., 2022; Zhang et al., 2022; Kirichenko et al., 2023; Moayeri et al., 2023; Deng et al., 2023).

## 7 CONCLUSION

**Summary.** We have shown that, in practice, extracting domain knowledge and information regarding the presence and nature of spurious features is often difficult. This renders all existing techniques that show promise in overcoming spurious correlations as ineffective. We also discover that spurious correlations are formed due to only a small fraction of all samples containing the spurious feature and develop a novel data pruning technique that overcomes spurious correlations by pruning a small subset of the training data that contains these samples. Finally, we show that such data pruning attains state-of-the-art performance on standard benchmarks where information regarding spurious features is easily available.

**Outlook.** With current practice of training models on increasing large datasets, it has become critical to develop techniques that identify samples that can cause these models to malfunction without requiring human intervention. In the future, we will further develop this work to take into account other failure modes that are commonly exhibited by these deep neural networks.

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

# A APPENDIX

## A.1 TRAINING DETAILS

**CIFAR-10S.** We use the ResNet20 implementation from Liu et al. (2019) that we train for 160 epochs. The network is optimized using SGD with an initial learning rate 1e-1 and weight decay 1e-4. The learning rate drops to 1e-2 and 1e-3 at epochs 80 and 120 respectively. We maintain a batch size of 64. Sample difficulty is computed after the 10th epoch.

**CelebA.** We use an ImageNet pre-trained ResNet-50 from PyTorch (Paszke et al., 2019) that we train for 25 epochs. The network is optimized using SGD with a static learning rate 1e-3 and weight decay 1e-4. We maintain a batch size of 64. Sample difficulty is computed after the 10th epoch.

**Hard Image-Net.** We use an ImageNet pre-trained ResNet-50 from PyTorch (Paszke et al., 2019) that we train for 50 epochs. The network is optimized using SGD with a static learning rate 1e-3 and weight decay 1e-4. We maintain a batch size of 128. Sample difficulty is computed after the 1st epoch.

**Waterbirds.** We use an ImageNet pre-trained ResNet-50 from PyTorch (Paszke et al., 2019) that we train for 100 epochs. The network is optimized using SGD with a static learning rate 1e-3 and weight decay 1e-3. We maintain a batch size of 128. Sample difficulty is computed after the 1st epoch.

**MultiNLI.** We use a pre-trained BERT model that we train for 20 epochs. The network is optimized using AdamW using a linearly decaying starting learning rate 2e-5. We maintain a batch size of 32. Sample difficulty is computed after the 5th epoch.

## A.2 ADDITIONAL EXPERIMENTAL DETAILS

**CIFAR-10S.** We follow a similar approach to Nagarajan et al. (2021) for adding a spurious line where pixel values for a vertical row of pixels in the middle of the first channel are set to the maximum possible value (255) before normalization and before any augmentations. We use the same augmentations generally used for training on the original CIFAR-10 Krizhevsky (2009).

**CelebA.** In this setting, we maintain 5000 Female Samples without Eyeglasses and 2500 Male samples with Eyeglasses and 2500 Male samples without Eyeglasses. Consistent with implementations in Sagawa et al. (2020a); Liu et al. (2021), we do not use any augmentations.

**Hard ImageNet.** In this setting, we maintain 59 Dog Sled samples with minimal spurious features and 100 Ski samples randomly drawn from the dataset. All remaining classes are maintained the same. We use the same augmentations used for training on ImageNet.

**Waterbirds.** We use the original Waterbirds setting commonly used in practice (Sagawa et al., 2020a; Liu et al., 2021; Zhang et al., 2022; Kirichenko et al., 2023). We use the augmentations used in Kirichenko et al. (2023) when training, which are similar to the augmentations used for training on ImageNet.

**MultiNLI.** We use the original MultiNLI setting commonly used in practice (Sagawa et al., 2020a; Liu et al., 2021; Kirichenko et al., 2023). Consistent with implementations in Sagawa et al. (2020a); Liu et al. (2021); Kirichenko et al. (2023), we do not use any augmentations.

## A.3 HARD IMAGENET: VERIFYING UNIDENTIFABILITY

We show that our Hard ImageNet setting is an unidentifiable setting by presenting information similar to Fig. 5a and Fig. 5b. But since there exist multiple spurious features, namely `Snow`, `People`, `Trees`, and `Hills`, we simply show that samples that contain these features and those that do not are scattered uniformly across the difficulty spectrum (Fig. 9).

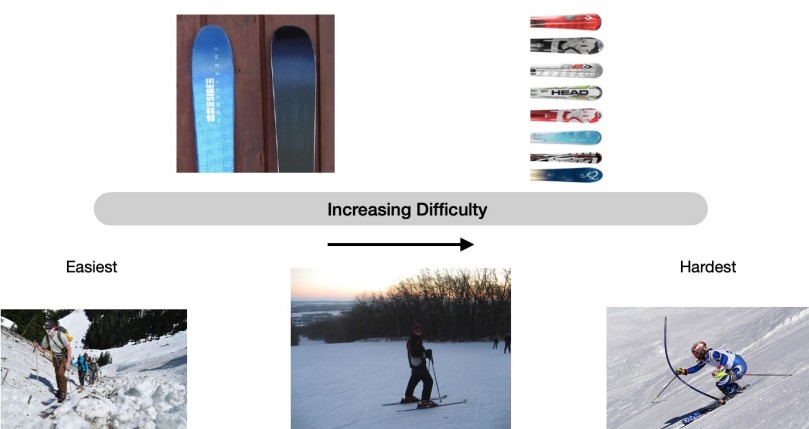

Figure 9: Training distribution variance based on strength of spurious feature in identifiable and unidentifiable settings.

### A.4    ADDITIONAL EXPERIMENTS WITH MULTINLI

In this section, we reinforce our claims and observations by re-conducting specific vision experiments in the language domain.

**Spurious Correlations are formed from a few key samples.** To show this, we perform the same experiment in Section 4, but instead of CIFAR-10S, we use the MultiNLI dataset. First, we remove all samples with negation words from the **training** data and then we compute the sample-wise difficulty scores as we do for CIFAR-10S in Section 4. We then create two settings: one where we introduce the spurious negation word "never" at the end of the 100 hardest input samples belonging to class 1 (`contradicts`) and another where we introduce the spurious negation word "never" at the end of the 100 easiest input samples belonging to class 1 (`contradicts`). We do the same to a set of test samples belonging to class 2 (`neutral with`) and class 3 (`entailed by`).

Consistent with the standard MultiNLI setting, we measure the degree of spurious feature reliance through Worst Group Accuracy (accuracy of the set of test samples of class 2 or class 3 with the spurious feature).

We observe that WGA is significantly worse when the word "never" occurs in the hardest samples vs. the easiest samples during training.

Introducing the spurious feature in easiest 100 samples: WGA = 55.22%

Introducing the spurious feature in hardest 100 samples: WGA = 1.04%

Additionally, we note that there are 191,504 training samples in this setting. There are 57,498 samples belonging to the `contradicts` class. We introduce the spurious feature in only 100 samples of the `contradicts` class (0.17% of samples within the class, 0.0522% of all samples in the training set.) We also observe that in a setting with no spurious features during training, Worst Group Accuracy is 67.42%.

This experiment reinforces the claim that samples with hard core features are primary contributors to spurious correlations and that samples with simple core features do not contribute to spurious correlations.

**Excluding a few key samples during training severs spurious correlations.**   In the MultiNLI setting, we observe that pruning the samples with hard core features and spurious features attains high worst group accuracy. On the other hand, excluding samples with easy core features and spurious features does not improve worst group accuracy. Note that we do not perform any hyperparameter tuning.

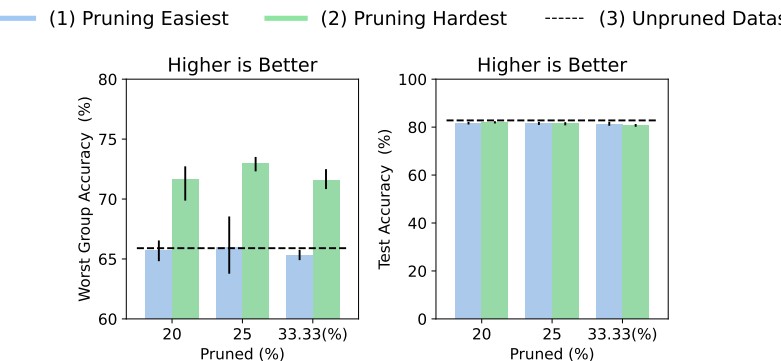

Figure 10: Excluding a fraction of all samples with hard core features and spurious features mitigates spurious correlations in the MultiNLI setting. This is indicated by high Worst Group Accuracies.

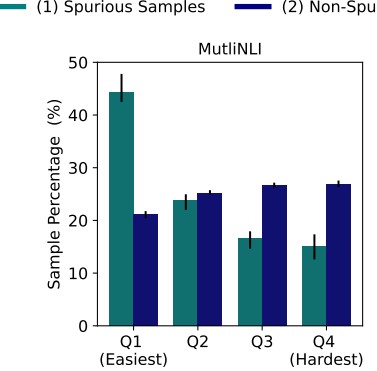

Figure 11: Training distribution variance based on strength of spurious feature in identifiable and unidentifiable settings. Grouped by Quartiles, sorted by difficulty.

**Distribution of the MultiNLI dataset.** We show the distribution of the MultiNLI dataset, an extension of Fig. 5b. Q1 in this setting contains almost half of all samples with spurious features while Q4 only contains 15%. This shows that samples with spurious features are not uniformly distributed when viewed through the lens of sample difficulty. This is in contrast to the CelebA setting (Unidentifiable), in which samples with spurious features are uniformly distributed (Fig. 5b (right)).

