# OpenReview forum: "Severing Spurious Correlations with Data Pruning"
_ICLR.cc/2025/Conference — ICLR 2025 Spotlight_

### Official Review · Reviewer_n9k8 · 2024-10-26

**Soundness:** 3
**Presentation:** 4
**Contribution:** 3
**Rating:** 8
**Confidence:** 4

**Summary:**

This paper empirically investigates the phenomenon of spurious correlations being learned by deep learning models.

The authors are mainly interested in the setting where the spurious signals are "weak", in the sense that are not easy to be detected and removed.

In this setting, the authors propose a data pruning approach to limit the effects of the spurious correlations.

The method consists in looking at the (few) training points that are more difficult to classify during training. These points are most likely making the model overfit patterns that are not necessarily causal to the right label, and are therefore a major cause to the learning of spurious patterns. This is confirmed experimentally.

**Strengths:**

The paper is very well written, it is clear, the approach followed by the authors is logically sound, and I personally enjoyed reading it. While I am not certain that it is true that all previous work adresses cases where the spurious signal is stronger than the core features, the contribution seems indeed novel, and the results are reasonable and well explained.

The paper is on point. It proposes mainly one single method, and the writing is structured in a way that the reader can digest the phenomenology before being given the final algorithm / approach.

**Weaknesses:**

It is sometimes not obvious what the authors mean by "strong" or "weak". While providing precise definitions is beyond the purposes of this work, some examples during the introduction could facilitate the reading. For example, I was confused in the paragraph at line 201, where the strength of the signal is defined both in terms of the geometry of the pattern and it's frequency in the data. These two aspects are fundamentally different, and putting them altogether might not result in the best model to investigate this problem...

The reason behing I do not give a higher schore is that some of the results are not entirely surprising. The phenomenlogy described by Figure 2 is interesting, but at the same time - to my understanding - in line with what we expect from the influence of individual samples to the final parameters of the model (see, e.g., [1, 2]). Nevertheless, this might be a new perspective in the community of spurious features / correlations, and I therefore recommend acceptance.

[1] https://arxiv.org/pdf/1906.05271
[2] https://proceedings.neurips.cc/paper_files/paper/2020/file/1e14bfe2714193e7af5abc64ecbd6b46-Paper.pdf

**Questions:**

There should be a typo in line 162 "as shown in..."

I am still confused about the setting used to obtain Figure 3. And what do the authors mean by "polynomial growth" in this case? What does it mean to "scale up the difficulty of the core features of the 100 saples"?

I am also confused by Section 6.3 - What do the authors mean by "group labels" in line 457? Can the authors provide a brief and self contained description of the point that wants to be raised in this section (while I believe this is not the central part of the work)?

---

> ### Author Response · Authors · 2024-11-18
> **Response to Reviewer n9k8 (Part 1)**
>
> We thank the reviewer for their comments. We are happy to hear that they found our work to be very well written and glad that they enjoyed reading our work. We have addressed their comments below:
>
> **It is sometimes not obvious what the authors mean by "strong" or "weak". While providing precise definitions is beyond the purposes of this work, some examples during the introduction could facilitate the reading. For example, I was confused in the paragraph at line 201, where the strength of the signal is defined both in terms of the geometry of the pattern and its frequency in the data. These two aspects are fundamentally different, and putting them altogether might not result in the best model to investigate this problem...**
>
> Thank you for this comment. The strength of the spurious signal indicates the ease of learning the spurious feature. When the strength of the spurious signal is strong, spurious features are learned easily and spurious correlations are formed easily. In all literature concerning spurious correlations, the strength of the spurious signal is primarily determined by the following three factors: (1) Proportion (or Frequency) of training samples containing the spurious feature (Sagawa et. al. 2020 ICML, Shah et. al. NeurIPS 2020, Kirichenko et. al. 2023 ICLR), (2) Area Occupied and Position (if it is centered or not) in the training sample (Moayeri et. al. 2022 NeurIPS) and (3) The amount of noise in the signal (Sagawa et. al. 2020 ICML, Ye et. al. 2023 AISTATS). A feature which is present in a large portion of all training samples, occupies a lot of area, is centered, and has little to no variance, has a very strong signal. On the other hand, a feature which is present in a small portion of all training samples, occupies little area, and has a lot of noise/variance, has a very weak signal. For instance, in Section 2, to reduce the strength of the spurious signal, we reduce the proportion of training samples that contain the spurious feature. We have added this to lines 128-136 of the revised text.
>
> We ensure that both aspects (geometry and frequency) are not modified in the same experiment. So for instance, in Section 5, the strength of the spurious signal for CIFAR-10S is varied by only changing the geometry of the spurious feature. In Section 6, the strength of the spurious signal for CIFAR-10S is varied by modifying the proportion of samples containing the presence of the spurious feature. We show that these experiments can be replicated by performing the other modification. Below, we present the results in Section 6 (Figure 5) for CIFAR-10S across three seeds by doing both: varying the geometry (or increasing the area occupied compared to the unidentifiable setting) and varying the proportion (increasing the proportion/frequency of samples containing the spurious feature compared to the unidentifiable setting):
>
> Identifiable Setting distribution by varying geometry (Spurious Samples only):
>
> Q1(Easiest): 53.2%, Q2: 24.26%, Q3: 16.13%, Q4 (Hardest): 6.39%
>
> Identifiable Setting distribution by varying proportion (Spurious Samples only, already in the paper):
>
> Q1(Easiest): 49.93%, Q2: 38.68%, Q3: 8.78%, Q4 (Hardest): 2.6%
>
> Unidentifiable Setting distribution (Spurious Samples only, already in the paper):
>
> Q1(Easiest): 30.93%, Q2: 22.26%, Q3: 23.73%, Q4 (Hardest): 23.06%
>
> We observe that in both Identifiable settings, Q1 contains most of the samples with spurious features while Q4 contains few samples with spurious samples.
>
> In other words, these modifications are interchangeable as they ultimately influence the same property: The strength of the spurious signal.
>
> \
> \
> (Shah et. al. 2020 NeurIPS) “The Pitfalls of Simplicity Bias in Neural Networks,” NeurIPS, 2020.
>
> (Sagawa et. al. 2020 ICML) “An Investigation of Why Overparameterization Exacerbates Spurious Correlations,” ICML, 2020.
>
> (Kirichenko et. al. 2023 ICLR) “Last Layer Re-training is Sufficient for Robustness to Spurious Correlations,” ICLR, 2023.
>
> (Moayeri et. al. 2022 NeurIPS) “Hard imagenet: Segmentations for objects with strong spurious cues”, NeurIPS, 2022.
>
> (Ye et. al. 2023 AISTATS) “Freeze then Train: Towards Provable Representation Learning under Spurious Correlations and Feature Noise”, AISTATS, 2023.
>
> \
> **There should be a typo in line 162 "as shown in..."**
>
> Thank you for pointing this out. We agree that there is a typo in line 162 and we fix the reference.

---

> ### Author Response · Authors · 2024-11-18
> **Response to Reviewer n9k8 (Part 2)**
>
> **"Scaling up the difficulty of the core features of the 100 samples."**
>
> Apologies for the confusion. In the CIFAR-10S experiments in Section 5, we take 100 samples to which we add synthetic spurious features and we keep changing the 100 samples like a 100-sample-sized sliding window to probe into. By scaling up the difficulty of the core features of the 100 samples, we simply increase the difficulty of the 100 samples into which we introduce the spurious features. We first sort the samples by difficulty. For results in Figure 2, we simply introduce spurious features in the easiest 100 and the hardest 100 samples. For results in Figure 3, we consider all unique 100 sample subsets by sliding a window of sample size 100 across the sorted samples list. Since there are 5000 total samples in the class, we have 50 different subsets in which spurious features are introduced. We will be more than happy to provide any further explanation.
>
> As we increase the difficulty of the 100 samples into which we introduce spurious features, we see that spurious feature reliance exhibits polynomial growth instead of linear growth with increasing difficulty.
>
> **What do the authors mean by “group labels” in line 457?**
>
> We sincerely apologize for not including a definition of group labels. We follow the literature concerning spurious features/correlations, where group labels commonly refer to labels that indicate the presence or absence of spurious features in each training sample within a class.
>
> Thus, within a class, samples with the spurious feature would have a group label = 1 and samples without the spurious feature would have a group label = 0. Note that this is different and independent from class labels in classification tasks. We make this point clearer in the paper. In identifiable settings (as in the existing literature) where the strength of the spurious signal is relatively stronger, it is trivial to identify group labels, as is shown in Section 3 (Figure 1 (b) (Right)).
>
>
>
> **Can the authors provide a brief and self contained description of the point that wants to be raised in this section (while I believe this is not the central part of the work)?**
>
> Our work is primarily interested in novel settings where spurious signals are weak, in that the signals are not easily identifiable, as you have described in sentence 2 of your review.
>
> However, in Section 6.3, we show that even in previously studied settings where the spurious signals are “strong” (identifiable settings), simply pruning a few samples can yield state-of-the-art results. While not the primary focus of our paper, we believe this finding is very important. Current techniques that attain good performance in these settings are extremely complex and computationally expensive. Our method is very simple to understand, takes only a few additional lines of code, is easy to reproduce and yields state-of-the-art performances, even on benchmarks that do not fit into the primary objectives of this paper.
>
> Please note that Waterbirds and MultiNLI are two of the most commonly studied benchmarks in literature concerning spurious correlations for Vision and Language tasks (Sagawa et. al. 2020 ICML, Kirichenko et. al. 2023 ICLR, Liu et. al. (ICML, 2021), Zhang et. al. (ICML, 2022), Ye et. al. 2023 AISTATS). Additionally, we highlight the robustness of our pruning technique by showing that pruning sparsities within a wide range can attain state-of-the-art or competitive performance on these benchmarks (Figure 8).
>
> We have improved the writing of this section thanks to your suggestions and questions. Please let us know if you have any concerns and we will be happy to address them.
>
> (Liu et. al. ICML, 2021) “Just Train Twice: Improving Group Robustness without Training Group Information,” ICML, 2021.
>
> (Zhang et. al. ICML, 2022) “Correct-n-Contrast: A Contrastive Approach for Improving Robustness to Spurious Correlations,” ICML, 2022.

---

> > ### Comment · Reviewer_n9k8 · 2024-11-21
> >
> > I thank the authors for their careful rebuttal.
> >
> > I am satisfied with it, it clarified my previous questions, and I therefore confirm my score. I would maybe recommend to not use the term "polynomial" in Figure 3, used to indicate something that simply grows "super-linearly". This behaviour is not proven to be anything fundamental accross datasets, so I wouldn't stress it that much during the narrative.
> >
> > On a separate matter, I also invite Reviewer 7BQb to reconsider their score, which I believe to not be in line with the value of this work.

---

> ### Author Response · Authors · 2024-11-21
> **Response to Reviewer n9k8 (Part 3)**
>
> We thank the reviewer for their thoughtful comment. We agree with their statement and have made the required changes to the revised text. We believe their comments and feedback have helped improve the quality of our work.
>
> We would also like to thank them for encouraging another peer reviewer, Reviewer 7BQb, to reconsider their score considering our work’s value. We hope that this discussion will lead to fair assessment of our work. Thank you again.

---

### Official Review · Reviewer_7BQb · 2024-10-28

**Soundness:** 2
**Presentation:** 2
**Contribution:** 2
**Rating:** 5
**Confidence:** 4

**Summary:**

This paper study the spurious feature associated with the trained model, focusing on how this correlation is formed during the training process and proposing a data pruning approach to mitigate spurious correlation. Specifically, the authors find that spurious correlation is caused by a few samples that are hard to learn and contains spurious features. As a result, this paper proposes to remove those data points from the training dataset to improve model performance.

**Strengths:**

1. The paper organization is clear and easy to follow.
2. The problem of identifying and mitigating spurious correlation is critial.
3. This paper considers a challenging scenario where spurious features have weak signals and therefore are difficult to be detected.

**Weaknesses:**

**Major Concerns:**

1. Lack of rigorous formulation and solution. Throughout this paper, there is no single equation or definition that clearly states the problem and the proposed solution. Key concepts that heavily mentioned in the paper, such as spurious correlation, core/invariant features, simple/hard features, are not well-defined. For a more readable paper, the authors are encouraged to (1) state a self-contained problem with properly defined concepts (2) write a pesudocode for the proposed data pruning algorithm (3) disclose complete experimental details including but not limited to training/test dataset processing procedure, models, calculation of evaluation metric (which is not well-defined as well), and baseline methods.

2. Lack of clarity in findings. Most figures and tables are not self-explained and neither explained by their titles. As a result, it is confusing how do they support the claims in the main text. This problem is aggravated due to the previous point. For example, I did not understand the message in Figure 2, as concepts like "easy/hard" are undefined, and authors do not explain how they perform the experiments, e.g., how to add spurious features into the training data and how to calculate the misclassification rates. The authors should review all their findings and claims and rewrite their evidence to be more supportive and convincing.

**Questions:**

Please find in the weakness section above.

---

> ### Author Response · Authors · 2024-11-18
> **Response to Reviewer 7BQb (Part 1)**
>
> We thank the reviewer for their comments. We have addressed all your concerns by providing all relevant definitions, detailed figure captions, technical details for the proposed approach and **additional** training details. Note that in addition to providing additional training details, we have also provided our code through an anonymous github repository to ensure easy reproducibility (Link: https://github.com/Anon-ICLRDP/ICLR_DP). We have included all these components in our paper but we provide them below for your convenience:
>
> **Key Concept Definitions:**
>
> **Lines 82-88 in revised text:**
> Consistent with past literature, we study the supervised classification setting where $S = {\{(x_i, y_i)}\}^N_{i=1}$ denotes the training dataset of size $N$ and network is trained to learn a mapping between $x_i$ (input) and $y_i$ (class label) using empirical risk minimization (Vapnik, 1998). Every training sample $s \in S$ contains a core feature ($c_i$) that represents its class ($y_i$). A fraction of all samples within a class contain the spurious feature $(a_i)$ associated with that class. **Core (or invariant) features** are causal to the class label $y_i$ and are fully predictive of the task, as they are present in all samples. **Spurious features** are not causal to the class labels and are partially predictive of the task, as they are present in only a fraction of all samples of a class.
>
> Past literature has found that during training, deep networks choose to rely on spurious features over core/invariant features if the spurious features are easier to learn than the core features. They form a correlation between these spurious features and ground truth labels. Such correlations are called spurious correlations.
>
> **Lines 90-92 in revised text:**
>
> **Spurious Correlations:** The correlation a network forms between spurious features and class labels. Such correlations are undesirable as they are not causal to class labels and can disappear during testing or become associated with a different task, causing these networks to malfunction.
>
>
> **Lines 128-136 in revised text:**
>
> **Feature Difficulty:** Consistent with deep learning literature (specifically, those works concerned with spurious correlations), difficulty of learning a feature is determined by the following three factors: (1) Proportion (or Frequency) of training samples containing the spurious feature (Sagawa et. al. 2020 ICML, Shah et. al. NeurIPS 2020, Kirichenko et. al. 2023 ICLR), (2) Area Occupied and Position (if it is centered or not) in the training sample (Moayeri et. al. 2022 NeurIPS) and (3) The amount of noise in the signal (Sagawa et. al. 2020 ICML, Ye et. al. 2023 AISTATS). A feature which is present in a large portion of all training samples, occupies a lot of area, is centered, and has little to no variance, is easy to learn. On the other hand, a feature which is present in a small portion of all training samples, occupies little area, is not centered, and has a lot of noise/variance, is hard to learn.
>
> (Shah et. al. 2020 NeurIPS) “The Pitfalls of Simplicity Bias in Neural Networks,” NeurIPS, 2020.
>
> (Sagawa et. al. 2020 ICML) “An Investigation of Why Overparameterization Exacerbates Spurious Correlations,” ICML, 2020.
>
> (Kirichenko et. al. 2023 ICLR) “Last Layer Re-training is Sufficient for Robustness to Spurious Correlations,” ICLR, 2023.
>
> (Moayeri et. al. 2022 NeurIPS) “Hard imagenet: Segmentations for objects with strong spurious cues”, NeurIPS, 2022.
>
> (Ye et. al. 2023 AISTATS) “Freeze then Train: Towards Provable Representation Learning under Spurious Correlations and Feature Noise”, AISTATS, 2023.
>
> **Problem Statement and Pseudocode:**
>
> **Lines 201-202 in revised text:**
>
> **Problem Statement:** How does one sever spurious correlations in settings where attaining spurious information is difficult or impossible?
>
>
> **Lines 290-294** in revised text:
>
> **Pseudocode:**
>
> 1) Train the network on the task for n epochs, where n << t and t is the total number of epochs in the training schedule.
> 2) Compute sample-wise difficulty scores as $|| p(w,x) - y||_2$ , where $p(w,x)$ is the probability distribution given by the network for sample $x$, $w$ denotes the network parameters after the nth epoch and $y$ is the one hot encoding of the ground truth value.
> 3) Prune samples with high difficulty scores.
> 4) Train a new network on the pruned dataset for t epochs.

---

> ### Author Response · Authors · 2024-11-18
> **Response to Reviewer 7BQb (Part 2)**
>
> **Clarifying Figure 2**: Please note that we have expanded on the figure caption and mentioned figure titles in the main text for better readability: (Lines 174-177 in text, with additional references in Section 5). For the reviewer, we provide a simple explanation below:
>
> The figures “Easiest” refers to injecting synthetic spurious features in the easiest 100 samples while the figures “Hardest” refers to injecting synthetic spurious features in the hardest 100 samples.
>
> **Explanation of evaluation metrics:** Reflecting on your comment, we added a more detailed explanation of the evaluation metrics used in our paper. Additional sentences: Lines 261-264 in text.
>
> Current practice in deep learning utilizes Worst-Group Accuracy (WGA) to assess the degree of spurious feature reliance in binary classification tasks. WGA computes the accuracy of test samples that contain the spurious feature associated with a different class during training. While suitable for simple binary classification tasks, WGA becomes insufficient to assess the reliance on spurious features in settings with multiple classes. This is because WGA cannot differentiate between loss in test accuracy due to spurious correlations, or due to lack of learnability of invariant correlations stemming from limited capacity or insufficient training data. In such settings, we measure the degree of spurious feature reliance through Spurious Misclassifications, i.e. the percentage of samples of one class (c1) containing the spurious feature of another class (c2) that are misclassified as (c2) during testing. Lower Worst Group Accuracy indicates heavy reliance on spurious correlations while high worst group accuracy indicates little to no reliance on spurious correlations. A high number of Spurious Misclassifications indicates heavy reliance on spurious correlations while a low number Spurious Misclassifications indicate little to no reliance on spurious correlations.
>
> **Experimental Details:**
>
> All experimental details are provided in the Appendix (Sections A.1 and A.2). We have already provided the models used and included all baseline methods in our evaluation. We provide additional experimental details within the same sections.
>
>
>
> **CIFAR-10S**. We use the ResNet20 implementation from Liu et. al. (ICLR, 2019) that we train for 160 epochs. The network is optimized using SGD with an initial learning rate 1e-1 and weight decay 1e-4. The learning rate drops to 1e-2 and 1e-3 at epochs 80 and 120 respectively. We maintain a batch size of 64. Sample difficulty is computed after the 10th epoch.
>
> **CelebA**.  We use an ImageNet pre-trained ResNet-50 from PyTorch Paszke et. al. (Neurips, 2019) that we train for 25 epochs. The network is optimized using SGD with a static learning rate 1e-3 and weight decay 1e-4. We maintain a batch size of 64. Sample difficulty is computed after the 10th epoch.
>
> **Hard Image-Net**.  We use an ImageNet pre-trained ResNet-50 from PyTorch Paszke et. al. (Neurips, 2019) that we train for 50 epochs. The network is optimized using SGD with a static learning rate 1e-3 and weight decay 1e-4. We maintain a batch size of 128. Sample difficulty is computed after the 1st epoch.
>
> **Waterbirds**.  We use an ImageNet pre-trained ResNet-50 from PyTorch Paszke et. al. (Neurips, 2019) that we train for 100 epochs. The network is optimized using SGD with a static learning rate 1e-3 and weight decay 1e-3. We maintain a batch size of 128. Sample difficulty is computed after the 1st epoch.
>
> **MultiNLI**. We use a pre-trained BERT model that we train for 20 epochs. The network is optimized using AdamW using a linearly decaying starting learning rate 2e-5. We maintain a batch size of 32. Sample difficulty is computed after the 5th epoch.

---

> ### Author Response · Authors · 2024-11-18
> **Response to Reviewer 7BQb (Part 3)**
>
> **Additional Experimental Details:**
>
> **CIFAR-10S.** We follow a similar approach to Nagarajan et. al. (ICLR, 2021) for adding a spurious line where pixel values for a vertical row of pixels in the middle of the first input channel are set to the maximum possible value (255) before normalization and before any augmentations. We use the same augmentations used for training on the original CIFAR-10.
>
> **CelebA.** In this setting, we maintain 5000 Female Samples without Eyeglasses and 2500 Male samples with Eyeglasses and 2500 Male samples without Eyeglasses. Consistent with the implementation in Sagawa et. al. (ICLR, 2019), Liu et. al. (ICML, 2021), we do not use any augmentations.
>
> **Hard ImageNet.**  In this setting, we maintain 58 Dog Sled samples with minimal spurious features and 100 Ski samples randomly drawn from the dataset. All remaining classes are maintained the same. We use the same augmentations used for training on ImageNet.
>
> **Waterbirds.**  We use the original Waterbirds setting commonly used in practice (Sagawa et. al. (ICLR, 2019), Liu et. al. (ICML, 2021), Zhang et. al. (ICML, 2022), Kirichenko et. al. (ICLR, 2023)). We use the augmentations used in Kirichenko et. al. (ICLR, 2023) when training, which are similar to the augmentations used for training on ImageNet.
>
> **MultiNLI.**   We use the original MultiNLI setting used in practice (Sagawa et. al. (ICLR, 2019), Liu et. al. (ICML, 2021), Kirichenko et. al. (ICLR, 2023)). Consistent with the implementation in Sagawa et. al. (ICLR, 2019), Liu et. al. (ICML, 2021), Kirichenko et. al. (ICLR, 2023)), we do not use any augmentations.
>
> (Nagarajan et. al. ICLR, 2021) “Understanding the failure modes of out-of-distribution generalization,” ICLR, 2021.
>
> (Shah et. al. 2020 NeurIPS) “The Pitfalls of Simplicity Bias in Neural Networks,” NeurIPS, 2020.
>
> (Sagawa et. al. 2020 ICML) “An Investigation of Why Overparameterization Exacerbates Spurious Correlations,” ICML, 2020.
>
> (Sagawa et. al. 2019 ICLR) “Distributionally Robust Neural Networks For Group Shifts: On the Importance of Regularization for Worst-Case Generalization,” ICLR, 2019
>
> (Kirichenko et. al. 2023 ICLR) “Last Layer Re-training is Sufficient for Robustness to Spurious Correlations,” ICLR, 2023.
>
> Liu et. al. (ICML, 2021) “Just Train Twice: Improving Group Robustness without Training Group Information,” ICML, 2021.
>
> Zhang et. al. (ICML, 2022) “Correct-n-Contrast: A Contrastive Approach for Improving Robustness to Spurious Correlations,” ICML, 2022.

---

> > ### Comment · Reviewer_7BQb · 2024-11-21
> >
> > I thank the authors for their detailed responses. The revised manuscript greatly enhances its readability and I have a better understanding on the main contribution of this paper.
> >
> > I would like to raise my score to 5, and am willing to increase if the following questions are also addressed:
> >
> > (1) The "strength" of a feature can be understood as its magnitude like the strength of signal/noise. It seems the strength in this paper actually represent its frequency. Is it correct? If so, I would suggest to replace strength as frequency.
> >
> > (2) While I agree with most observations and the proposed method for unidentifiable case, could the authors explain how to prune in the identifiable setting? Given that the authors claim "Yang et al. (2024) show that in settings where the strength of the spurious signal is significantly greater than the strength of the invariant signal, it is possible to identify which samples contain spurious features in them and which ones do not", how to identify spurious samples as stated in "simply pruning those spurious samples containing the hardest core features". Also, what does that mean by "we work with group labels as is done in ..."?
> >
> > (3) Can authors clarify the rationale behind "the presence of strong spurious information enables the network to understand samples with hard core features better"?
> >
> > (4) Suppose a sample has a hard invariant feature and an easy spurious feature, should it be easy or hard to learn (small or large training error)? My understanding is that the sample diffculty is estimated by the training error, in this case, it is unclear how to identify "spurious samples containing the hardest core features".

---

> > > ### Author Response · Authors · 2024-11-22
> > > **Response to Reviewer 7BQb (Part 4)**
> > >
> > > We sincerely thank the reviewer for taking the time to go through the revised version of our paper. We address their follow-up concerns below:
> > >
> > > **(1) The "strength" of a feature can be understood as its magnitude like the strength of signal/noise. It seems the strength in this paper actually represent its frequency. Is it correct? If so, I would suggest to replace strength as frequency.**
> > >
> > > In our paper, the strength of spurious signals is **varied** by two main factors: **frequency** (Sec. 3 & 6) and **area** (Sec. 4.) This is consistent with existing literature (Lines 128 - 136 in our paper).
> > >
> > > Below, we categorize sections of our paper where strength is varied by frequency or area:
> > >
> > > **By Frequency:**
> > >
> > > Section 3: Observing increased drops in Worst Group Accuracy (Female Samples with the Spurious Feature) by increasing the frequency of Male Samples with the Spurious Feature.
> > >
> > > Section 6: In the synthetic CIFAR-10S setting, we vary the strength of the spurious features (between identifiable and unidentifiable settings) by varying the frequency.
> > >
> > > **By Area:**
> > >
> > > Section 4: In the synthetic CIFAR-10S setting, we vary the strength of the three spurious features (S1, S2, and S3) by varying the amount of area that they take up in the image. S1 takes up the least amount of area and causes the least amount of spurious misclassifications (less spurious feature reliance). S3 takes up the most amount of area and causes the most amount of spurious misclassifications (more spurious feature reliance). S2 is in between S1 and S3. We observe that introduction of S3 (largest area) causes the most number of spurious misclassifications while introduction of S1 (smallest area) causes the least number of spurious misclassifications (Figure 2). **Please note that we do not vary the frequency of samples containing the spurious feature in the three settings. In other words, the same number (= 100) of samples contain spurious features occupying different areas.**
> > >
> > > It is important to note the two attributes (frequency and area) can be used interchangeably to alter the strength of the spurious signal. Below, we present the distribution results in Section 6 (Figure 5 (a)) for CIFAR-10S across three seeds by modifying both attributes: varying the proportion (or increasing the proportion/frequency of samples containing the spurious feature compared to the unidentifiable setting) and varying the area (or increasing the area occupied compared to the unidentifiable setting).
> > >
> > > Identifiable Setting distribution by varying area **only** (Spurious Samples only):
> > >
> > >
> > > | Q1 (Easiest) | Q2 | Q3 | Q4 (Hardest) |
> > > | ------- | ------- | ------- | ------- |
> > > |53.2% | 24.26% | 16.13% | 6.39% |
> > >
> > > Identifiable Setting distribution by varying proportion **only** (Spurious Samples only, already in the paper):
> > >
> > >
> > > | Q1 (Easiest) | Q2 | Q3 | Q4 (Hardest) |
> > > | ------- | ------- | ------- | ------- |
> > > | 49.93% | 38.68% | 8.78% | 2.6%|
> > >
> > > Unidentifiable Setting distribution (Spurious Samples only, already in the paper):
> > >
> > >
> > > | Q1 (Easiest) | Q2 | Q3 | Q4 (Hardest) |
> > > | ------- | ------- | ------- | ------- |
> > > | 30.93% | 22.26% | 23.73% | 23.06% |
> > >
> > > In the two identifiable settings (varying the area vs. varying the frequency), the distributions of spurious samples are similar, where Q1 contains most of the samples with spurious features while Q4 contains very few samples with spurious samples.
> > >
> > >
> > > While in our experiments, these are the only two factors that we vary, most benchmarks studied in spurious correlations literature exhibit a combination of the factors mentioned in our paper (Lines 128 - 136 in our paper). Consider the popular Waterbirds dataset, where spurious features occupy a lot of area: Water and Land backgrounds in the image. It is unlikely that the network will form spurious correlations if the background only spans a few pixels and contains a lot of noise, even if one were to maintain the frequency of samples containing the spurious features. This claim is supported by our experiments in Section 4, where if the spurious feature covers less area (S1), the network has minimal reliance on spurious features even though the same number of samples contain the spurious features across the three settings: S1, S2, and S3.

---

> > > > ### Author Response · Authors · 2024-11-22
> > > > **Response to Reviewer 7BQb (Part 5)**
> > > >
> > > > **(2) While I agree with most observations and the proposed method for unidentifiable case, could the authors explain how to prune in the identifiable setting? Given that the authors claim "Yang et al. (2024) show that in settings where the strength of the spurious signal is significantly greater than the strength of the invariant signal, it is possible to identify which samples contain spurious features in them and which ones do not", how to identify spurious samples as stated in "simply pruning those spurious samples containing the hardest core features". Also, what does that mean by "we work with group labels as is done in ..."?**
> > > >
> > > > Identifiable settings have been extensively studied in literature (Sagawa et. al. 2020 ICML, 2019 ICLR, Kirichenko 2023 ICLR, to list a few). (Sohoni et. al. 2020 NeurIPS, Liu et. al. 2021 ICML, Zhang et. al. 2022 ICML, Ahmed et. al. 2021 ICLR, Creager et. al. ICML 2021, Yang et. al. 2024 AISTATS) found that in these identifiable settings, deep neural networks form a very strong reliance on spurious features. Thus, through a network's learned (biased) representations, they were able to identify which samples within a class contain spurious features and which ones do not. One way is to cluster the network’s representations of its training samples. Since the network relies very strongly on spurious features and ignores invariant features, clusters are formed based on the presence/absence of spurious features instead of class labels. Identifying which cluster contains the spurious feature associated with that class can be done by clustering at specific points in the training schedule or by observing the margin with which samples within a cluster are classified. Another popular way to identify samples containing spurious features in identifiable settings is to simply train with high regularization, as we have already shown in Section 3 (Figure 1(b) Right). Samples with spurious features are correctly classified whereas samples without spurious features are incorrectly classified.
> > > >
> > > > Since it has been established that it is possible to identify samples containing spurious features in identifiable/popular settings, most seminal works (Sagawa et. al. 2019, ICLR, Kirichenko 2023 ICLR, Deng et. al. 2023 NeurIPS) simply make use of information regarding the sample-wise presence of spurious features. For simplicity and to reduce the number of components in our paper, we do the same. Additionally, in Table 1 of our text, we primarily compare our results to those that directly make use of information regarding the sample-wise presence of spurious features.
> > > >
> > > > Building on the previous paragraph, we explain group labels:
> > > >
> > > > In most literature concerning spurious features/correlations, group labels commonly refer to labels that indicate the presence or absence of spurious features in each training sample within a class. Thus, within a class, samples with the spurious feature would have a group label = 1 and samples without the spurious feature would have a group label = 0. Note that this is different and independent from class labels in classification tasks. We have made this point clearer in the revised text (Lines 463-464). In identifiable settings (as in the existing literature) where the strength of the spurious signal is relatively stronger, it is trivial to identify group labels, as we have explained above.
> > > >
> > > > Finally and more importantly, we would like to point out that the primary purpose of our work is to study and tackle novel settings where spurious features/information is unidentifiable and propose novel insights (Spurious correlations are formed from a handful of all samples containing spurious features). The purpose of Section 6.3 is to show that in settings where spurious features/signals are strong and identifiable, simply pruning a few samples can yield state-of-the-art results. While not the main focus of our paper, we believe this finding is very important. Current techniques that attain good performance in these settings are extremely complex and computationally expensive. Our method is very simple to understand, takes only a few additional lines of code, is easy to reproduce and yields state-of-the-art performances, even on benchmarks that do not fit into the primary objectives of this paper. The robustness of our findings, and by extension, their importance, is further highlighted by observing that pruning sparsities of a wide range can attain state-of-the-art or competitive performance on these benchmarks (Figure 8).

---

> ### Author Response · Authors · 2024-11-22
> **Response to Reviewer 7BQb (Part 6)**
>
> **(3) Can authors clarify the rationale behind "the presence of strong spurious information enables the network to understand samples with hard core features better"?**
>
>
> By the statement highlighted by the reviewer, we mean that samples with hard core features incur a low training error as the network relies on strong spurious features to reduce training error for that sample. Thanks to your comment, we think this sentence might create future confusion and thus, rephrase it in the revised text.
> “The presence of strong spurious information enables the network to have lower training error for samples with hard core features + spurious features.”
>
> To support the statement, we kindly refer the reviewer to Figure 5, where in the presence of strong spurious information (identifiable settings), most samples with spurious features have low training error. Additionally, to further reinforce the validity of this claim, we ran the following experiment:
>
> We perform the same experiment in Section 4, where we first compute the training error of all samples when trained on clean CIFAR-10. We then introduce the spurious feature S3 into 100 samples with the highest training error, which gives us the results in Figure 2 (c) and (d). We additionally compare the average training error of the **same** 100 samples with and without the spurious features across three seeds:
>
> Without spurious feature S3: 1.30
>
> With spurious feature S3: 0.68
>
> We observe that by simply introducing strong spurious features, samples with hard core features (those that incur a high training error initially) no longer have a high training error.
>
> \
> \
> (Sagawa et. al. 2020) “An Investigation of Why Overparameterization Exacerbates Spurious Correlations,” ICML, 2020.
>
> (Sagawa et. al. 2019 ICLR) “Distributionally Robust Neural Networks For Group Shifts: On the Importance of Regularization for Worst-Case Generalization,” ICLR, 2019.
>
> (Kirichenko et. al. 2023) “Last Layer Re-training is Sufficient for Robustness to Spurious Correlations,” ICLR, 2023.
>
> (Sohoni et. al. 2020) “No subclass left behind: Fine-grained robustness in coarse-grained classification problems,” NeurIPS 2020.
>
> (Liu et. al. ICML, 2021) “Just Train Twice: Improving Group Robustness without Training Group Information,” ICML, 2021.
>
> (Zhang et. al. ICML, 2022) “Correct-n-Contrast: A Contrastive Approach for Improving Robustness to Spurious Correlations,” ICML, 2022.
>
> (Ahmed et. al. ICLR 2021) “Systematic generalisation with group invariant predictions,” ICLR 2021.
>
> (Creager et. al. 2021) “Environment inference for invariant learning,” ICML 2021.
>
> (Yang 2024) “Identifying spurious biases early in training through the lens of simplicity bias,” AISTATS 2024.
>
> (Deng et. al. 2023) “Robust learning with progressive data expansion against spurious correlation,” NeurIPS 2023.
>
> \
> \
> **(4) Suppose a sample has a hard invariant feature and an easy spurious feature, should it be easy or hard to learn (small or large training error)? My understanding is that the sample diffculty is estimated by the training error, in this case, it is unclear how to identify "spurious samples containing the hardest core features".**
>
>
> When a sample has a hard invariant feature and an easy spurious feature, it becomes easier to learn (low training error) than when it only has the invariant feature. In identifiable settings (which we understand is the setting you are referring to), samples with hard core features + spurious features become easier to learn than samples without spurious features. This is why **we cannot simply prune the hardest samples in the identifiable settings** (Lines 456 - 457).
>
> In identifiable settings, however, it is very easy to identify which samples contain spurious features. In other words, it is very easy to identify group labels, by clustering for instance, as explained in question 2. In such settings, since we are aware of which samples contain spurious features, we prune those samples with spurious features that have a higher training error. These would include samples containing easy spurious features + hard core features versus those that contain samples containing easy spurious features + easy core features.
>
> We would like to re-iterate that finding group labels in identifiable settings is trivial and we follow other seminal works that directly use group labels to mitigate spurious correlations. Note that we also compare our results with techniques that directly use group labels to mitigate spurious correlations.
>
>
> \
> \
> We will be more than happy to address any further concerns that the reviewer might have. Thank you for taking the time and effort to go through the revised version of our paper and suggesting thoughtful comments to improve the quality of the paper.

---

> > ### Author Response · Authors · 2024-11-25
> >
> > Dear Reviewer 7BQb,
> >
> > Kindly let us know if we have resolved the 4 follow-up concerns. If not, we will be happy to address any remaining concerns. Thank you.

---

> > > ### Comment · Reviewer_7BQb · 2024-11-25
> > >
> > > I appreciate the response from the authors. My questions 2, 3 and 4 are addressed in the response. Nevertheless, the question 1 remains.
> > >
> > > The authors state that the strength of a feature includes both its frequency and area. Those concepts are actually not well-defined. For example, the eye-glasses is considered the spurious feature in the CelebA dataset. While eyeglasses can have different shapes, colors, and sizes, red eyeglasses shall be conceivably easier to identify than a transparent-frame. I would like to express two points: the "magnitude" is clearly an important aspect of the feature strength, and "feature" itself is not well-defined, thus making it vague in what scenarios the conclusions of this work will hold. The latter concern agrees with the comment of Reviewer X6Fn on the generalizability.

---

> ### Author Response · Authors · 2024-11-26
> **Response to Reviewer 7BQb (Part 7)**
>
> We thank the reviewer for their thoughts and comments.
>
> We would like to clarify that in our responses, we only state that we vary frequency and area alone. If the reviewer has concerns regarding these factors, we refer them to papers (already cited in our text, Lines 127 - 135) that study their role in the strength of (spurious/core) features - Sagawa et. al. 2020 ICML, Shah et. al. 2020 NeurIPS, Moayeri et. al. 2022 NeurIPS, etc. We simply follow them. There are additional factors that can influence the strength of a feature that we do not vary/modify in our experiments. In the definition provided in the text, we also mention noise in the signal as a factor that influences the strength of a feature. To the best of our knowledge, current literature studies only these three factors extensively and this is why we only include these in our definitions.
>
> Most works in spurious correlations consider the simple setting where samples belonging to a class contain one core feature and may contain one spurious feature associated with that class (Sagawa et. al. 2020 ICML, Liu et. al. 2021 ICML, Zhang et. al. 2022 ICML, to list a few). In Vision settings, features are described as objects. In Language settings, features are described as words or sentences.
>
> The definitions we provide are consistent with what is currently known and accepted in the literature. We do not claim to encompass all possible factors and it is for this reason that we do not provide numeric values regarding the strengths of features in the 5 settings studied. We clarify that this is beyond the scope of this work.
>
> As most works in spurious correlations literature, we start with settings where spurious correlations are formed (identifiable or unidentifiable/novel) and propose solutions to tackle them, while providing novel insights and attaining SOTA on previously studied settings.
>
> \
> (Sagawa et. al. 2020 ICML) “An Investigation of Why Overparameterization Exacerbates Spurious Correlations,” ICML, 2020.
>
> (Shah et. al. 2020 NeurIPS) “The Pitfalls of Simplicity Bias in Neural Networks,” NeurIPS, 2020.
>
> (Moayeri et. al. 2022 NeurIPS) “Hard imagenet: Segmentations for objects with strong spurious cues”, NeurIPS, 2022.
>
> (Liu et. al. ICML, 2021) “Just Train Twice: Improving Group Robustness without Training Group Information,” ICML, 2021.
>
> (Zhang et. al. ICML, 2022) “Correct-n-Contrast: A Contrastive Approach for Improving Robustness to Spurious Correlations,” ICML, 2022.
>
> \
> Kindly let us know if there are any remaining concerns and we will be happy to address them.

---

> > ### Author Response · Authors · 2024-11-29
> >
> > Dear Reviewer 7BQb,
> >
> > Kindly let us know if you are satisfied with our answer to your last question.
> >
> > To re-iterate, we only claim to **vary** two factors in our experiments: frequency and area. There are many factors that can impact the strength of a feature, as discussed in literature (Sagawa et. al. 2020 ICML, Shah et. al. 2020 NeurIPS, Moayeri et. al. 2022 NeurIPS) and mentioned in our paper (Lines 127-135). Note that we only present those factors that are extensively discussed in literature. Discovering and considering all factors that may impact the strength of a feature/signal is beyond the scope of this work.
> >
> > To address generalizability concerns (as you have cited Reviewer X6Fn’s comment), we would like to emphasize that we consider most existing benchmarks in literature, propose new ones and present experiments across different architectures and domains (Vision and Language). We would also like to point out that we have satisfied Reviewer X6Fn’s generalizability concerns.
> >
> > If the reviewer still has any concerns, we present below many key vision experiments in the paper in the language setting to further reinforce the robustness of our insights and observations (in addition to the MultiNLI results already present in the paper.)

---

> > > ### Author Response · Authors · 2024-11-29
> > >
> > > **Insights 1 and 2:**
> > >
> > > **Samples with simple core features do not contribute to spurious correlations. Samples with hard core features are primary contributors to spurious correlations.**
> > >
> > > To show this, we perform the same experiment in Section 4, but instead of CIFAR-10S, we use the MultiNLI dataset. First, we remove all samples with negation words from the **training** data and then we compute the sample-wise difficulty scores as we do for CIFAR-10S in Section 4. We then create two settings: one where we introduce the spurious negation word “never” at the end of the 100 hardest input samples belonging to class 1 (**contradicts**) and another where we introduce the spurious negation word “never” at the end of the 100 easiest input samples belonging to class 1 (**contradicts**). We do the same to a set of test samples belonging to class 2 (**neutral with**) and class 3 (**entailed by**).
> > >
> > > Consistent with the standard MultiNLI setting, we measure the degree of spurious feature reliance through Worst Group Accuracy (accuracy of the set of test samples of class 2 or class 3 with the spurious feature).
> > >
> > > We observe that WGA is significantly worse when the word “never” occurs in the hardest samples vs. the easiest samples during training.
> > >
> > > Introducing spurious feature in easiest 100 samples: WGA = **55.22%**
> > >
> > > Introducing spurious feature in hardest 100 samples: WGA = **1.04%**
> > >
> > > A higher WGA indicates low reliance on spurious features.
> > >
> > > The gap in worst group accuracy is 54.18%. Note that the number of samples containing the spurious feature is the same in both settings (= 100).
> > >
> > > Additionally, we note that there are 191,504 training samples in this setting. There are 57,498 samples belonging to the **contradicts** class. We introduce the spurious feature in only 100 samples of the **contradicts** class (0.17% of samples within the class, 0.0522% of all samples in the training set.) We also observe that in a setting with no spurious features during training, Worst Group Accuracy is 67.42%.
> > >
> > > Simply varying which 100 samples contain the spurious negation word “never” has such a **huge** impact on Worst Group Accuracy. This finding is extremely insightful, novel and is consistent with the results observed in Section 4 (Figure 2) of our paper.
> > >
> > > This experiment reinforces the claim that samples with hard core features are primary contributors to spurious correlations and that samples with simple core features do not contribute to spurious correlations.
> > >
> > > **Insight 3:**
> > >
> > > **Excluding a few key samples during training severs spurious correlations.**
> > >
> > > For this, we simply show the results in Figure 4 but for the MultiNLI dataset. Note: Due to computational constraints, we only show three pruning sparsities.
> > >
> > > Worst Group Accuracy:
> > >
> > > | Prune %   | 20%| 25%| 33.33%|
> > > | -------- | ------- | ------- | ------- |
> > > |Pruning Easiest| 66.81| 65.59| 65.33|
> > > |Pruning Hardest| 72.21| 73.17| 76.05|
> > >
> > > Kindly note that the model attains 65.9% Worst Group Accuracy on the original, unpruned dataset.
> > >
> > > We refer the Reviewer to Fig. 10 in the appendix for a better understanding.
> > >
> > > **Insight 4:**
> > >
> > > **Spurious feature strength created a specific distribution of the training data.**
> > >
> > > To show this, we use a smaller subset of the MultiNLI dataset and vary the strength of the spurious signal by varying the proportion of samples containing the spurious feature.
> > >
> > > Distribution of samples with spurious features in Identifiable setting:
> > >
> > > | Q1 (Easiest) | Q2 | Q3 | Q4 (Hardest) |
> > > | ------- | ------- | ------- | ------- |
> > > |57.4% | 24.3% | 11.5% | 6.8% |
> > >
> > > Distribution of samples with spurious features in Unidentifiable setting:
> > >
> > > | Q1 (Easiest) | Q2 | Q3 | Q4 (Hardest) |
> > > | ------- | ------- | ------- | ------- |
> > > |28% | 21% | 24% | 27% |
> > >
> > > To confirm that the unidentifiable setting still causes the network to rely on spurious correlations, we create another setting where we remove all samples with spurious features and compare the Worst Group Accuracies below:
> > >
> > > Unidentifiable setting: 64.89%
> > > No samples with spurious features setting: 70.73%
> > >
> > > We observe that in the setting with no samples containing the spurious features, Worst Group Accuracy is higher, indicating that the unidentifiable setting still causes the network to rely on spurious correlations but the samples containing spurious features are uniformly distributed.
> > >
> > > ---
> > > \
> > > Kindly let us know if there are any remaining concerns and we will be happy to address them.

---

> ### Author Response · Authors · 2024-12-02
>
> Dear Reviewer 7BQb,
>
> We hope our latest responses have helped address your concern. Kindly let us know if your concern still remains.
>
> Thank you.

---

### Official Review · Reviewer_eCbW · 2024-11-03

**Soundness:** 2
**Presentation:** 3
**Contribution:** 2
**Rating:** 5
**Confidence:** 3

**Summary:**

The paper explores how deep neural networks often rely on spurious correlations present in training data, which can lead to performance drop under distributional shifts. The authors highlight a novel setting where spurious signals are weaker, making their identification challenging. They propose a new data pruning technique that selectively removes some training samples that contribute significantly to the formation of spurious correlations. This technique operates without requiring detailed information on the nature or presence of spurious features. The authors demonstrate that their approach achieves state-of-the-art performance on both standard and challenging benchmarks, including scenarios where spurious features are identifiable and unidentifiable.

**Strengths:**

1) The paper addresses a scenario where the strength of the spurious signal is relatively weaker and thus it is difficult to detect any spurious informationhat spurious signals compared to other works  where the strength of the spurious signal is significantly greater
than that of the core.
2) The paper provides a thorough experimental design that spans both identifiable and unidentifiable spurious feature scenarios. The use of multiple datasets (e.g., CIFAR-10S, CelebA, Hard ImageNet, Waterbirds, MultiNLI) showcases the method's robustness.
3) The paper is well-organized, detailing the rationale behind the proposed method, experimental setup, and results.

**Weaknesses:**

1) The paper could explore more thoroughly the practicality of applying the proposed pruning method to large datasets. Specifically,even though the proposed method shows promise for datasets of moderate size, an assessment of its computational cost and efficiency on large and real world data would be beneficial.
2) The approach relies on assessing sample difficulty as a proxy for contribution to spurious correlations. Clarifying the robustness of this metric under different training regimes (e.g., varied architectures or optimization strategies) could strengthen the generalizability of the findings.
3) Although the paper discusses state-of-the-art methods, further comparative analysis with recently emerging pruning and robust training techniques that do not rely on explicit spurious feature identification would be helpful.

**Questions:**

1) How sensitive is the pruning method to changes in hyperparameters?
2) The robustness of sample difficulty estimation across different model architectures is not clear at the current version of the paper. Would it be possible to add some results or explanations for that?
3) Table 1 shows that some SOTa methods achieve better results. Could you please give some explanation on that? Also,
4) could you please open source the code to enhance the reproducibility?
Suggestions for Improvement:
1) It would be helpful to include a section that discusses the potential negative impacts or limitations of the method, such as the risk of pruning samples that are informative but rare.
2)Extending the empirical analysis to more complex and real-world datasets with non-synthetic spurious correlations could further validate the applicability of the method.

---

> ### Author Response · Authors · 2024-11-18
> **Response to Reviewer eCbW (Part 1)**
>
> We thank the reviewer for their comments. We address their concerns below:
>
> **Weakness 1: The paper could explore more thoroughly the practicality of applying the proposed pruning method to large datasets. Specifically, even though the proposed method shows promise for datasets of moderate size, an assessment of its computational cost and efficiency on large and real-world data would be beneficial.**
>
>
> We thank the reviewer for their comment. We would like to emphasize that while efficiency was not the primary objective of our work, data pruning is an efficient solution compared to other techniques as one only does standard training on a pruned (thus, smaller) dataset. This is in contrast to other techniques that do sample upweighting or representational alignment, which significantly increases the training time over just standard training. Additionally, we emphasize that Hard ImageNet, Waterbirds, MultiNLI, and CelebA studied in our paper are real-world datasets with realistic/real-world spurious features, and we intentionally utilized CIFAR-10S to better understand how varying the strength of (synthetic) spurious features impacts generalizability and training distribution (Lines 219-222, Lines 301-317, Figure 5(a).) Regarding the matter of scale, ImageNet is the only dataset that contains samples on a scale that is different from CIFAR-10 but it is not studied in the context of spurious correlations. To the best of our knowledge, there is no dataset at that scale that is studied in the context of spurious correlations.
>
> **Weakness 2: The approach relies on assessing sample difficulty as a proxy for contribution to spurious correlations. Clarifying the robustness of this metric under different training regimes (e.g., varied architectures or optimization strategies) could strengthen the generalizability of the findings.**
>
>
> We show that the computation of these ranks and subsequent pruning are robust even across architectures and different optimization strategies/hyperparameters. Below, we present the worst Group Accuracy for the CelebA Setting (Figure 4) for different architectures and hyperparameters for the same Prune Percentage (%). We will include these results in the revised text:
>
>
> ResNet18:
> | Prune %   | 10%| 25%| 40%| 50%| 75%| 90%| 95%| 97%|
> | -------- | ------- | ------- | ------- | ------- | ------- | ------- | ------- | ------- |
> |Pruning Easiest| 26.33| 24.56| 23.63| 22.85| 22.43| 17.1| 26.33| 35.84|
> |Pruning Hardest| 50.18| 73.67| 68.49| 74.31| 84.17| 89.35| 80.2| 83.32|
>
> VGG16:
>
> | Prune %   | 10%| 25%| 40%| 50%| 75%| 90%| 95%| 97%|
> | -------- | ------- | ------- | ------- | ------- | ------- | ------- | ------- | ------- |
> | Pruning Easiest| 33.45| 28.62| 28.34| 29.88| 36.39| 60.53| 71.73| 77.96|
> |Pruning Hardest| 37.58| 54.09| 68.93| 71.59| 82.16| 85.86| 86.49| 85.23|
>
> Learning Rate = 0.01:
> | Prune %   | 10%| 25%| 40%| 50%| 75%| 90%| 95%| 97%|
> | -------- | ------- | ------- | ------- | ------- | ------- | ------- | ------- | ------- |
> |Pruning Easiest| 18.98| 15.76| 12.11| 22.76| 18.63| 30.11| 14.36| 51.96|
> |Pruning Hardest| 49.72| 48.67| 66.88| 74.44| 82.21| 78.29| 87.32| 85.99|
>
> Learning Rate = 0.0001:
> | Prune %   | 10%| 25%| 40%| 50%| 75%| 90%| 95%| 97%|
> | -------- | ------- | ------- | ------- | ------- | ------- | ------- | ------- | ------- |
> |Pruning Easiest| 19.49| 16.68| 17.8| 15.55| 14.5| 18.23| 35.05| 51.37|
> |Pruning Hardest| 45.74| 59.32| 68.9| 73.47| 79.38| 85.22| 87.61| 86.49|
>
>
>
> Weight Decay = 0.001:
> | Prune %   | 10%| 25%| 40%| 50%| 75%| 90%| 95%| 97%|
> | -------- | ------- | ------- | ------- | ------- | ------- | ------- | ------- | ------- |
> |Pruning Easiest| 18.06| 15.64| 19.86| 16.26| 13.98| 25.05| 18.13| 22.98|
> |Pruning Hardest| 42.63| 62.63| 70.31| 75.22| 82.7| 87.82| 88.3| 86.23|
>
> Weight Decay = 0.01:
> | Prune %   | 10%| 25%| 40%| 50%| 75%| 90%| 95%| 97%|
> | -------- | ------- | ------- | ------- | ------- | ------- | ------- | ------- | ------- |
> |Pruning Easiest| 12.74| 15.56| 13.75| 17.44| 14.54| 36.03| 21.64| 27.06|
> |Pruning Hardest| 38.57| 53.11| 57.31| 73.81| 83.43| 90.3| 87.26| 87.7|
>
> **Weakness 3: Although the paper discusses state-of-the-art methods, further comparative analysis with recently emerging pruning and robust training techniques that do not rely on explicit spurious feature identification would be helpful.**
>
> To the best of our knowledge, there do not exist any promising pruning or robust training techniques that do not rely on explicit spurious feature identifications. We will be happy to compare our approach with techniques based on the reviewer’s recommendation.

---

> ### Author Response · Authors · 2024-11-18
> **Response to Reviewer eCbW (Part 2)**
>
> **Question 1: How sensitive is the pruning method to changes in hyperparameters?
> Question 2: The robustness of sample difficulty estimation across different model architectures is not clear at the current version of the paper. Would it be possible to add some results or explanations for that?**
>
> Current experiments are run on ResNet-50 and BERT, which is a transformer-based model. We also include experimental results on VGGNets, smaller ResNets and different hyperparameters in response to Weakness 2.
>
> **Question 3: Table 1 shows that some SOTa methods achieve better results. Could you please give some explanation on that? Also, could you please open source the code to enhance the reproducibility?**
>
> Only one existing SOTA method achieves better results (Worst-Group Accuracy) on only one dataset (MultiNLI). While we do not have an explanation for this, it is important to note that results on identifiable benchmarks are not the primary focus of the paper. The primary focus of our paper is on novel settings where spurious information is unidentifiable. Additionally, current techniques that attain good performance in these settings are extremely complex and computationally expensive. Our method is very simple to understand and implement -  requiring only a few additional lines of code, is easy to reproduce, and yields state-of-the-art performance, even on benchmarks that do not fit into the primary objectives of this paper.
>
> Sure, to enhance reproducibility, we have added an anonymized repository for the experimental results presented in this paper, and we will open-source it upon this paper’s acceptance. Kindly let us know if you have any difficulties executing this code and reproducing the results. Link: https://github.com/Anon-ICLRDP/ICLR_DP
>
>
> **Question 4: Suggestions for Improvement:**
>
> **1) It would be helpful to include a section that discusses the potential negative impacts or limitations of the method, such as the risk of pruning samples that are informative but rare.**
>
> We thank the reviewer for this suggestion. We believe that it is important to include such limitations. However, in our extensive empirical evaluation across multiple different datasets, we see little to no reduction in testing accuracy when pruning these key samples because they generally do not contribute much to generalizability (Lines 366 - 368, Figure 4 (Right), Figure 6 (Right), Figure 7 (Right), Table 1 (Mean Accuracy)). If one were to continue to prune more of the harder samples, test accuracy would drop (Sorscher et. al., 2022 NeurIPS). However, based on our current observations, it is clear that the amount of data needed to be pruned to severe spurious correlations is less than the amount of data needed to observe noticeable drops in test accuracy. We will be happy to perform any further analysis to assess the potential impacts or limitations of the method based on the reviewer’s recommendation.
>
> (Sorscher et. al., 2022 NeurIPS) Sorscher, Ben, et al. "Beyond neural scaling laws: beating power law scaling via data pruning.”, 2022 NeurIPS.
>
> **2) Extending the empirical analysis to more complex and real-world datasets with non-synthetic spurious correlations could further validate the applicability of the method.**
>
> We would like to emphasize that across the 5 datasets that we study (HardImageNet, CelebA, Waterbirds, MultiNLI and CIFAR-10S), 4 of them are real-world datasets containing real-world spurious features and correlations. We intentionally included one dataset containing synthetic spurious features (CIFAR-10S) as it allows us to vary the strength of the spurious feature relative to the core, invariant feature. All of the findings from the one synthetic dataset translate perfectly to the four real-world datasets.
>
> To the best of our knowledge, there do not exist any datasets utilized in literature for spurious correlations that could offer additional or better insights. However, we will be more than happy to include results from any datasets that the reviewer would suggest that we include in our analysis.

---

> > ### Author Response · Authors · 2024-11-25
> >
> > Dear Reviewer eCbW,
> >
> > Kindly let us know if we have resolved your concerns. We have provided results across different architectures and hyperparameters, have released our code through an anonymous github link, and have provided explanations for the remaining questions.
> >
> > We will be happy to address any remaining concerns. Thank you.

---

> > > ### Author Response · Authors · 2024-11-29
> > >
> > > Dear Reviewer eCbW,
> > >
> > > As per your request, we have:
> > >
> > > 1) Open sourced the code.
> > > 2) Addressed your questions and concerns.
> > > 3) Provided additional experimental results across architectures and hyperparameters.
> > >
> > > ---
> > > \
> > > To further reinforce the generalizability of our insights and observations, we have re-created certain critical Vision experiments in the paper for Language tasks as well (in addition to the original MultiNLI experiments already present in the paper.)
> > >
> > > **Insights 1 and 2:**
> > >
> > > **Samples with simple core features do not contribute to spurious correlations. Samples with hard core features are primary contributors to spurious correlations.**
> > >
> > > To show this, we perform the same experiment in Section 4, but instead of CIFAR-10S, we use the MultiNLI dataset. First, we remove all samples with negation words from the **training** data and then we compute the sample-wise difficulty scores as we do for CIFAR-10S in Section 4. We then create two settings: one where we introduce the spurious negation word “never” at the end of the 100 hardest input samples belonging to class 1 (**contradicts**) and another where we introduce the spurious negation word “never” at the end of the 100 easiest input samples belonging to class 1 (**contradicts**). We do the same to a set of test samples belonging to class 2 (**neutral with**) and class 3 (**entailed by**).
> > >
> > > Consistent with the standard MultiNLI setting, we measure the degree of spurious feature reliance through Worst Group Accuracy (accuracy of the set of test samples of class 2 or class 3 with the spurious feature).
> > >
> > > We observe that WGA is significantly worse when the word “never” occurs in the hardest samples vs. the easiest samples during training.
> > >
> > > Introducing spurious feature in easiest 100 samples: WGA = **55.22%**
> > >
> > > Introducing spurious feature in hardest 100 samples: WGA = **1.04%**
> > >
> > > A higher WGA indicates low reliance on spurious features.
> > >
> > > The gap in worst group accuracy is 54.18%. Note that the number of samples containing the spurious feature is the same in both settings (= 100).
> > >
> > > Additionally, we note that there are 191,504 training samples in this setting. There are 57,498 samples belonging to the **contradicts** class. We introduce the spurious feature in only 100 samples of the **contradicts** class (0.17% of samples within the class, 0.0522% of all samples in the training set.) We also observe that in a setting with no spurious features during training, Worst Group Accuracy is 67.42%.
> > >
> > > Simply varying which 100 samples contain the spurious negation word “never” has such a **huge** impact on Worst Group Accuracy. This finding is extremely insightful, novel and is consistent with the results observed in Section 4 (Figure 2) of our paper.
> > >
> > > This experiment reinforces the claim that samples with hard core features are primary contributors to spurious correlations and that samples with simple core features do not contribute to spurious correlations.
> > >
> > > **Insight 3:**
> > >
> > > **Excluding a few key samples during training severs spurious correlations.**
> > >
> > > For this, we simply show the results in Figure 4 but for the MultiNLI dataset. Note: Due to computational constraints, we only show three pruning sparsities.
> > >
> > > Worst Group Accuracy:
> > >
> > > | Prune %   | 20%| 25%| 33.33%|
> > > | -------- | ------- | ------- | ------- |
> > > |Pruning Easiest| 66.81| 65.59| 65.33|
> > > |Pruning Hardest| 72.21| 73.17| 76.05|
> > >
> > > Kindly note that the model attains 65.9% Worst Group Accuracy on the original, unpruned dataset.
> > >
> > > We refer the Reviewer to Fig. 10 in the appendix for a better understanding.
> > >
> > > **Insight 4:**
> > >
> > > **Spurious feature strength created a specific distribution of the training data.**
> > >
> > > To show this, we use a smaller subset of the MultiNLI dataset and vary the strength of the spurious signal by varying the proportion of samples containing the spurious feature.
> > >
> > > Distribution of samples with spurious features in Identifiable setting:
> > >
> > > | Q1 (Easiest) | Q2 | Q3 | Q4 (Hardest) |
> > > | ------- | ------- | ------- | ------- |
> > > |57.4% | 24.3% | 11.5% | 6.8% |
> > >
> > > Distribution of samples with spurious features in Unidentifiable setting:
> > >
> > > | Q1 (Easiest) | Q2 | Q3 | Q4 (Hardest) |
> > > | ------- | ------- | ------- | ------- |
> > > |28% | 21% | 24% | 27% |
> > >
> > > To confirm that the unidentifiable setting still causes the network to rely on spurious correlations, we create another setting where we remove all samples with spurious features and compare the Worst Group Accuracies below:
> > >
> > > Unidentifiable setting: 64.89%
> > > No samples with spurious features setting: 70.73%
> > >
> > > We observe that in the setting with no samples containing the spurious features, Worst Group Accuracy is higher, indicating that the unidentifiable setting still causes the network to rely on spurious correlations but the samples containing spurious features are uniformly distributed.
> > >
> > > ---
> > > \
> > > Kindly let us know if there are any remaining concerns and we will be happy to address them.

---

> ### Author Response · Authors · 2024-12-02
>
> Dear Reviewer eCbW,
>
> It would be great if we could hear back from you so that we can know if we have addressed all of your concerns.
>
> Thank you.

---

### Official Review · Reviewer_X6Fn · 2024-11-05

**Soundness:** 3
**Presentation:** 4
**Contribution:** 4
**Rating:** 10
**Confidence:** 4

**Summary:**

This paper aims to mitigate the problem of spurious correlations in deep learning models. Through a sequence of simulation experiments, they authors discover that a small subset of training data containing “hard” core features is primarily responsible for the model learning spurious correlations, even when the spurious signal is weak. The authors then demonstrate through subsequent experiments that pruning this subset of samples effectively severs the link between spurious and core features.

**Strengths:**

This work represents an example of my favorite kind of work. A very important and easy-to-understand problem (spurious correlations learned by models that affect their generalizability) and a very intuitive solution, once the authors explain and demonstrate it. I also felt the work was well written, guided the reader through the gaps in the literature, and step by step demonstrated the veracity of their arguments with simple and compelling experiments.

**Weaknesses:**

Inductive Solution
There I assume many types of spurious correlations can be learned by a model. For example, it is will documted that outliers can overinfluence functional estimation, and create spurious corrleations. The proposed solution addresses only a specific type of spurious correlation (i.e., roughly speaking, where a certain class has an over-representation of a given feature that its not truly indicative of the class label), which arises under the particular conditions the authors generate (i.e., "the spurious feature takes the form of a line running through the center of the images", "images of men with glasses", etc. ), given the distributional properties of the datasets considered, and given deep learning architectures (e.g., mostly ResNet) considered. Moreover, their solution relies heavily on specific empirical observations about how this particular type of spurious correlation manifests and is distributed, given the particular generation process followed. For instance, the authors observe that in their simulations "in settings where the strength of the spurious signal is not significantly greater than the strength of the invariant signal...samples containing spurious features are uniformly distributed across the training distribution...[and] the presence of spurious features does not have a significant impact on the training distribution...[therefore] samples containing hard core features that also contain the spurious feature are primary contributors to the formation of spurious correlations". They therefore, conclude that "to mitigate spurious correlations, one would only have to prune the hardest samples in the training set, as this subset of the data would contain samples with spurious features that have hard core features.” This reasoning appears to be purely inductive and raises questions about the generalizability of the observations and subsequent solution. Specifically, I'd assume that the observations about the distribution of spurious signals and sample difficulty will hold across other types of spurious correlations, which may behave differently under varying dataset characteristics or model architectures.

Theoretical Justification/Generality
Without a theoretical foundation supporting the general applicability of the inducitve observations, it remains unclear whether the observations that lead to solution are universal and/or if the prunning method can serve as a general approach to mitigating spurious correlations in models. Therefore, the paper could be significantly improve with a theoretical justifications for the generality of the inductive observation. For example, 1) is the particular type of spurious correlation the authors consider representative of all spurious correlations, 2) does the distributional uniformality of samples containing features with spurious correlation across the training samples generalize across types of spurious correlations, 3) is the pruning of this particular type of subset as a solution to (any type of) spurious correlations, and 4) are these justifications architecture/data dependent.

Related, but unique, I think structuring the pruning solution as a formal model, statistical hypothesis test, etc. would strengthen the theoretical foundation of the proposed pruning technique, and again perhaps shed light on it's generality. A theoretical or statistical clarity on why the samples containing spurious correlation are distributed uniformaly (e.g., I wonder if this is a consequence of the inverse probability transform). A claim that spurious feature reliance experienceing polynomial growth with increasing sample difficulty, would be more trustworthy if supported by some theoretical formulation or model.

Pruning Consequences
While the authors demonstrate that prunning a particular type of subset of data points will reduce spurious correlations, there is no discussion of what are the consequences. I am inclined to beleive that throwing out data is going to have some sort of negative consequence, and therefore it important to know what trade-off is being made. This will allow a better comparison to other methods that do not simply prune data as well as enable practitioners to understand/determine if the cost of pruning data is worth the benefit in removing spurious correlations.

**Questions:**

My question(s) would simply be can the authors provide theory, models, statistical justification etc. (or fairly robust/general empirics) to address the listed weaknesses?

---

> ### Author Response · Authors · 2024-11-18
> **Response to Reviewer X6Fn (Part 1)**
>
> We thank the reviewer for their comments and detailed review. We are happy to hear that our work represents your favorite kind of work and that our paper is written well with compelling experiments.
>
> Before we address your questions, we would like to emphasize that the scope of our paper is to provide empirical evidence regarding the gaps in literature, novel insights regarding the behavior of deep neural networks in the presence of spurious correlations and the effectiveness of our proposed solution. To do so, we make sure to cover most benchmarks generally studied in spurious correlations literature (identifiable settings) while introducing new ones. To the best of our knowledge, there do not exist other benchmarks that may offer new insights but we will be more than happy to verify our claims on any other benchmarks based on your recommendation.
>
> **Question 1: Is the particular type of spurious correlation the authors consider representative of all spurious correlations?**
>
> Thank you for this question. The spurious correlations we study are consistent with all previous literature that studied spurious correlations, where spurious features are easier to learn than core features and are not fully predictive of the task. In settings where this does not hold (so for instance, settings where the spurious features are harder to learn than the core features), the network will choose to ignore these features (Shah et. al. 2020 NeurIPS) and so the spurious correlations present in the dataset are not learned. In other words, we are primarily concerned with settings where the network relies on spurious correlations present in the dataset and exclude those settings where spurious correlations are present in the dataset but are not learned. The latter has never been studied in deep learning literature. We are unaware of any other spurious correlations previously studied.
>
> Within the setting studied, we define two types of spurious correlations. One where the strength of the spurious signal is significantly greater than the core signal (identifiable) and another where the strength of the spurious signal is relatively weaker (unidentifiable, novel).
>
> Additionally, we would like to point out that not all benchmarks studied in this paper have spurious features that are over-represented in a certain class. In Section 2, we show that spurious correlations are formed even when 10% (or 50%) of all samples within a class contain the spurious feature.
>
> If the reviewer's concern was with respect to the kinds of spurious features studied, we would like to emphasize that our experiments consider benchmarks where spurious features are backgrounds (snow, water and land backgrounds), objects or words (eyeglasses, trees, negation words, etc.) or synthetic lines through which we believe we covered almost all commonly used benchmarks regarding spurious features.
>
> (Shah et. al. 2020 NeurIPS) “The Pitfalls of Simplicity Bias in Neural Networks,” NeurIPS, 2020.
>
> **Question 2: Does the distributional uniformality of samples containing features with spurious correlation across the training samples generalize across types of spurious correlations?**
>
> The answer is no. In Section 6.1, we show that in settings where the strength of the spurious signal is significantly greater than the strength of the core features (Identifiable Settings), samples containing spurious features are no longer uniformly distributed (Figure 5). The key attribute which determines the distribution is the strength of the spurious signal in the training set. In settings where the strength of the signal is significantly greater, however, it is trivial to identify the presence of spurious features in training samples.
>
> **Question 3: Is the pruning of this particular type of subset as a solution to (any type of) spurious correlations?**
>
> Yes. We are unaware of any other type of spurious correlation learned by deep neural networks and we have covered almost all existing benchmarks studied in literature. However, we are happy to verify our claims on any other benchmarks based on your recommendation.

---

> ### Author Response · Authors · 2024-11-18
> **Response to Reviewer X6Fn (Part 2)**
>
> **Question 4: Are these justifications architecture/data dependent?**
>
> No, the justifications are not architecture/data dependent. Our original experiments are conducted on ResNet-50 and Transformer based architectures like BERT. We test on 5 different datasets with different numbers of classes, input dimensions, difficulties, and spurious features. To further reinforce our empirical findings, we show that we obtain similar results with VGG16 and smaller ResNets like ResNet18. We also show that our results on the CelebA setting are consistent across different hyperparameters such as different learning rates and weight decays. We will include these results in the appendix.
>
> Worst Group Accuracy of Figure 4 for different architectures/hyperparameters for the same Prune Percentage (%):
>
> ResNet18:
> | Prune %   | 10%| 25%| 40%| 50%| 75%| 90%| 95%| 97%|
> | -------- | ------- | ------- | ------- | ------- | ------- | ------- | ------- | ------- |
> |Pruning Easiest| 26.33| 24.56| 23.63| 22.85| 22.43| 17.1| 26.33| 35.84|
> |Pruning Hardest| 50.18| 73.67| 68.49| 74.31| 84.17| 89.35| 80.2| 83.32|
>
> VGG16:
>
> | Prune %   | 10%| 25%| 40%| 50%| 75%| 90%| 95%| 97%|
> | -------- | ------- | ------- | ------- | ------- | ------- | ------- | ------- | ------- |
> | Pruning Easiest| 33.45| 28.62| 28.34| 29.88| 36.39| 60.53| 71.73| 77.96|
> |Pruning Hardest| 37.58| 54.09| 68.93| 71.59| 82.16| 85.86| 86.49| 85.23|
>
> Learning Rate = 0.01:
> | Prune %   | 10%| 25%| 40%| 50%| 75%| 90%| 95%| 97%|
> | -------- | ------- | ------- | ------- | ------- | ------- | ------- | ------- | ------- |
> |Pruning Easiest| 18.98| 15.76| 12.11| 22.76| 18.63| 30.11| 14.36| 51.96|
> |Pruning Hardest| 49.72| 48.67| 66.88| 74.44| 82.21| 78.29| 87.32| 85.99|
>
> Learning Rate = 0.0001:
> | Prune %   | 10%| 25%| 40%| 50%| 75%| 90%| 95%| 97%|
> | -------- | ------- | ------- | ------- | ------- | ------- | ------- | ------- | ------- |
> |Pruning Easiest| 19.49| 16.68| 17.8| 15.55| 14.5| 18.23| 35.05| 51.37|
> |Pruning Hardest| 45.74| 59.32| 68.9| 73.47| 79.38| 85.22| 87.61| 86.49|
>
>
>
> Weight Decay = 0.001:
> | Prune %   | 10%| 25%| 40%| 50%| 75%| 90%| 95%| 97%|
> | -------- | ------- | ------- | ------- | ------- | ------- | ------- | ------- | ------- |
> |Pruning Easiest| 18.06| 15.64| 19.86| 16.26| 13.98| 25.05| 18.13| 22.98|
> |Pruning Hardest| 42.63| 62.63| 70.31| 75.22| 82.7| 87.82| 88.3| 86.23|
>
> Weight Decay = 0.01:
> | Prune %   | 10%| 25%| 40%| 50%| 75%| 90%| 95%| 97%|
> | -------- | ------- | ------- | ------- | ------- | ------- | ------- | ------- | ------- |
> |Pruning Easiest| 12.74| 15.56| 13.75| 17.44| 14.54| 36.03| 21.64| 27.06|
> |Pruning Hardest| 38.57| 53.11| 57.31| 73.81| 83.43| 90.3| 87.26| 87.7|
>
>
> **Question 5: While the authors demonstrate that prunning a particular type of subset of data points will reduce spurious correlations, there is no discussion of what are the consequences. I am inclined to beleive that throwing out data is going to have some sort of negative consequence, and therefore it important to know what trade-off is being made. This will allow a better comparison to other methods that do not simply prune data as well as enable practitioners to understand/determine if the cost of pruning data is worth the benefit in removing spurious correlations.**
>
> We thank the reviewer for this suggestion. We also believe that it is important to include such limitations. However, in our extensive empirical evaluation across five different datasets with different numbers of classes, input dimensions, difficulties, and spurious features, we see little to no reduction in testing accuracy when pruning these key samples because they generally do not contribute much to generalizability (Lines 366 - 368, Figure 4 (Right), Figure 6 (Right), Figure 7 (Right), Table 1 (Mean Accuracy)). If one were to continue to prune more of the harder samples, test accuracy will drop (Sorscher et. al., 2022 NeurIPS). However, based on our current observations, it is evident that the amount of data needed to be pruned to severe spurious correlations is less than the amount of data needed to observe noticeable drops in test accuracy. We will be happy to perform any further analysis to assess the potential impacts or limitations of the method based on the reviewer’s recommendation.
>
> (Sorscher et. al., 2022 NeurIPS) "Beyond neural scaling laws: beating power law scaling via data pruning.”, 2022 NeurIPS.

---

> > ### Comment · Reviewer_X6Fn · 2024-11-24
> > **Meaning of spurious correlations and Causality**
> >
> > While I still very much like this work, I would like first to clarify my questions about the "representations of all spurious correlations" and then respectfully push back on what the authors argue they have proven with the work.
> >
> > **Question 1: Is the particular type of spurious correlation the authors consider representative of all spurious correlations?**
> > I thank the authors for the clarification, though I do want to be clear about the nature of my question and make a request, given their updated paper. For context, I have carefully read a handful of papers in the literature that the authors cite--i.e., Shah et al. 2020, Sagawa et al. (2020a), Kirichenko et al. (2022), Moayeri et al. (2023), Liu et al. (2021), Zhang et al. (2022), and accept they are being consistent with this literature, focused specifically on deep learning (image) models. The source of my original question is that generally speaking spurious correlations do not simply exist in this literature, and can occur by different mechanisms, as a simple example is in the context of linear regression, a single outlier can cause the estimated regression slope to be significantly different from zero. Moreover, [spurious correlations](https://en.wikipedia.org/wiki/Spurious_relationship) are generally considered relationships learned in data between $X$ and $Y, when $Y has no actual *causal* impact on $Y$. Causality is a very precise, well-defined, and well-studied concept across various disciplines, and it does not appear that the authors are truly attempting to operate in the context of causality. While I would not hold the authors responsible for the use of spurious correlations outside of the context of causality, I would like to recommend they amend the causality references they adopted (I believe in response to Reviwers 7BQb concerns): e.g.,
> >
> > > "Core (or invariant) features are causal to the class label $y_i$ and are
> > > fully predictive of the task, as they are present in all samples."
> >
> > I would instead recommend following the definition used in [Singhla and Feizi, 2022](https://arxiv.org/abs/2110.04301), as it seems to often be cited by others in the authors' literature, as they seem to work to precisely codify the concept, even for their MTurkers. Instead, for example, I believe the "present in all samples" component of the authors' criteria is not a requirement of core features (Liu et al. (2021) discuss background features that are not present in all samples) and seems like it could be true of a "spurious" feature for a given dataset.

---

> > ### Comment · Reviewer_X6Fn · 2024-11-24
> > **Implied Claims of Generalizability**
> >
> > **General Critique**
> > At the core of all my comments is a concern of generalizability, i.e., to what other contexts the observations and subsequent conclusions the authors make can be applied. As discussed above, the idea of a spurious correlation has a broader meaning beyond the context the authors are considering, I feel this is simply a matter of intention and terminology. Therefore, other than the above recommendations, I'm happy to limit ourselves to the class of spurious correlations from the authors' literature, and their intended scope:
> >
> > > "the scope of our paper is to provide empirical evidence regarding the gaps in the literature, novel insights regarding the behavior of deep neural networks in the presence of spurious correlations, and the effectiveness of our proposed solution."
> >
> > I completely understand this scope, but I do not understand how this scope and the empirical work the authors have done, support the stated contributions
> >
> > > **Contributions:**
> > > 1.  Identifying and targeting novel settings where obtaining spurious information is difficult/impossible and showing the failure of past techniques in these settings.
> > > 2.  Discovering that spurious correlations are primarily formed from a handful of all samples containing spurious features.
> > > 3.  Proposing a novel data pruning solution that severs spurious correlations in all novel settings while attaining state-of-the-art performances on previously studied settings.
> >
> > This scope can certainly (and seems to successfully) support Contribution 1 and consider a particular context of spurious correlation where other techniques have failed in the past. This is possible because to demonstrate failure, one simply needs to show examples of said failure. However, Contribution 2 and the first half of Contribution 3 are general claims about how spurious correlations manifest and subsequently the general efficacy of their proposed pruning solution. The space of ways deep learning models are being used in practice is massive, while the authors consider 5 image datasets where they either generate a particular type of feature to serve as a spurious correlation (e.g., line across the image) or rely on examples of previously studied, identified, or labeled spurious correlations. I completely understand the authors' choice to do so, and is in fact what I would likely do as well. I also understand that their literature has only provided this small set of labeled/benchmark data, so I agree they can claim to have obtained SOTA performance on previously studied settings, as this is something they can again demonstrate empirically. However, my challenge is that the paper (and the authors' responses to the review team) are written as if their observations about the spurious features they studied, the behavior of the spurious correlations induced by the features, and the response to the spurious correlation to their data pruning in their specific experiments are generally true of all spurious correlation learned from a "spurious" feature a deep learning model would use instead of the "core" feature.
> >
> > To make this point salient, it would need to be true that all these empirical behaviors present in their simulations are indicative of the set of all such possible spurious correlations learnable by DNNs. This is why I asked about theoretical justifications, the formality of the pruning procedure as a statistical test, etc. because these could then allow us to analyze the (probabilistic) conditions of the data and the spurious feature such that authors reported contributions 2 and 3, enabling (at least to some degree) is to understand the generalizability of the observations. Again, the authors' response, that their scope is to provide empirical evidence, is well taken, but then we don't know what behaviors are generalizable or to what degree. Will everything hold if I'm considering DNN with the potential to learn spurious correlation in other contexts: tabular data, textual data, or speech data; I could see arguments for and against the result porting over. Or given that the image datasets the authors use generally have high test accuracy, does it all hold when the underlying prediction task is more difficult? To be clear, I do not want to encourage (or even imply) the authors should engage in a "wac-a-mole" exercise, given the limited space they have to begin with, because these are just a few quick examples.  Instead, I hope that this helps elucidate the importance of the point: while I like the result and I think it's likely not limited to *just* the precise simulations the authors consider, I don't know how general it is. Moreover,  I feel the authors are making strong positive assertions about the behaviors of DNN models with spurious correlation and the power of their data pruning solution, well beyond what they can prove. And frankly, I don't think they need to because the observations are exciting on their own!

---

> > ### Comment · Reviewer_X6Fn · 2024-11-24
> > **Relevant Example**
> >
> > In closing, I'll highlight that the authors cited Shah et. al., 2020 in their response to my original review; this work provides a formal theory on when the network will (not) learn core (complex) vs spurious (simple) features, which then they expand on with simulations. Therefore when Shah et. al., 2020 make claims like "Neural networks exhibit simplicity bias" and "Extreme simplicity bias leads to non-robustness" it allows us to more deeply interrogate facets of their claims and understand their generalizability to settings they do not explicitly consider. If the authors of this present work were able to provide some theoretical results that even conjecture about the presence of (easy/hard) training samples with spurious features to the forming of spurious correlations and/or their removal on the accuracy, this would give way more credence to the claimed contributions and certainly would merit a higher score from me. If not, I would still like to congratulate the authors on a really interesting paper and strongly encourage them to reconsider what they feel they must claim as proven.

---

> ### Comment · Reviewer_X6Fn · 2024-11-23
> **Apologies on Late Discussion Engagment**
>
> I apologize to the authors and the review team for jumping into this discussion period a little later than others, I had an unexpectedly sick infant at home this week.

---

> > ### Author Response · Authors · 2024-11-23
> > **Summary of our response.**
> >
> > Dear Reviewer X6Fn,
> >
> > Thank you for your acknowledgment. We provide a condensed version of our response for your convenience:
> >
> > ---
> >
> >
> > **Point 1**
> >
> > The scope of our work is to provide empirical evidence regarding/addressing (1) gaps in literature, (2) novel insights about the behavior of deep neural networks in the presence of spurious correlations, and (3) the effectiveness of our proposed solution.
> >
> > To do so, we make sure to cover most benchmarks commonly studied in spurious correlations literature while introducing new ones.
> >
> > **Point 2**
> >
> > **Question 1: Is the particular type of spurious correlation the authors consider representative of all spurious correlations?**
> >
> > We cover all types of spurious correlations present in most existing benchmarks in literature concerning spurious correlations and propose new ones as well. We are unaware of any other benchmarks that may offer additional insights.
> >
> > **Point 3**
> >
> > **Question 2: Does the distributional uniformality of samples containing features with spurious correlation across the training samples generalize across types of spurious correlations?**
> >
> > No, in Section 6.1, we show that if the strength of the spurious signal is significantly greater (identifiable settings), the distributional uniformality no longer holds.
> >
> > **Point 4**
> >
> > **Question 3: Is the pruning of this particular type of subset as a solution to (any type of) spurious correlations?**
> >
> > Yes, we have shown that pruning a few key players severs spurious correlations in all types of spurious correlations studied in our paper, which, to the best of our knowledge, is representative of all spurious correlations that are currently known and studied.
> >
> > **Point 5**
> >
> > **Question 4: Are these justifications architecture/data dependent?**
> >
> > No. Our claims and observations hold for 5 datasets and different architectures (ResNets and Transformer based models, already in the paper) and we have also shown that such pruning is robust to changes in hyperparameters and additional architectures.
> >
> > **Point 6**
> >
> > **Question 5: Concerns regarding trade-offs**
> >
> > In our evaluation across 5 different datasets with very diverse characteristics, we see little to no reduction in testing accuracy. Based on our observations, it is evident that the amount of data needed to be pruned to sever spurious correlations is less than the amount needed to observe significant drops in test accuracy.
> >
> > ---
> >
> > We hope this helps. Thank you.

---

> ### Author Response · Authors · 2024-11-25
> **Response to Reviewer X6Fn (Part 3)**
>
> We thank the reviewer for their comments. We are glad that they like our work very much, believe that the observations are exciting and that our paper is really interesting. It is encouraging to see these comments. We try our best to address their follow-up concerns below:
>
> \
> **I would instead recommend following the definition used in Singhla and Feizi, 2022, as it seems to often be cited by others in the authors' literature, as they seem to work to precisely codify the concept, even for their MTurkers.**
>
> We thank the reviewer for this comment. We would like to clearly note that our settings cover **both Vision and Language**, so we cannot directly use the definition from Singhla and Feizi, 2022, because their definition is valid only for Vision settings (directly defined on visual features and objects). Also, please note that Singhla and Feizi, 2022 state that core attributes are the set of visual features that are **always** a part of the object definition. Additionally, in Liu et. al. 2021, there are no core features that are backgrounds. The only background features they consider are present in the Waterbirds task (Land/Water background) and these are spurious features, not core features. The core feature associated with each class is present in all samples of that class (e.g., all samples belonging to the Landbirds class contain landbirds (core feature) in them. Importantly, not all of the Landbird samples contain land backgrounds (spurious feature associated with that class). Some samples contain water backgrounds. (Section B (B.1) in the appendix). However, we are happy to alter the definitions to incorporate your comments as shown below:
>
> >Core (or invariant) features represent the class label $y_i$ and are semantically relevant to the task. They are also fully predictive of the task, as they are present in all samples. Spurious features do not represent the class labels and are semantically irrelevant to the task.
>
> We simply use the definitions provided by Izmailov et. al. 2022 NeurIPS and Kirichenko et. al. 2023 ICLR. We make these changes to the revised text.
>
> \
> Izmailov et. al. 2022 NeurIPS, On Feature Learning in the Presence of Spurious Correlations
>
> Kirichenko et. al. 2023 ICLR, Last Layer Re-Training is Sufficient for Robustness to Spurious Correlations
>
> \
> \
> While we soften our claims below, we would like to point out that our empirical analysis covers 5 diverse datasets and different architectures. Additionally, we emphasize that the papers the reviewer references (Liu et al. (2021), Kirichenko et. al. (2023), Moayeri et al. (2023)) propose solutions and insights with empirical evidence alone. However, subsequent works then used their insights to create theories and techniques (e.g. Ye et. al. 2023 AISTATS). Not only does our work make use of the benchmarks mentioned in these works, but we also propose new settings where these techniques are not suitable. We believe pushing the boundary on what is currently known is a good step toward covering all spurious correlations in the future or creating a better understanding of the behavior of deep networks. For instance, the papers cited above do not cover all spurious correlations, as we show that their methods are not suitable in the proposed settings. However, the insights and contributions of these papers were critical in moving the field forward. While we do not prove our claims theoretically, we strongly believe that the insights and contributions of our paper have their own novel value and are important for future research.
>
> That being said, we are happy to soften the claims by making changes to the contributions in the following manner.
>
> Changes in contributions listed in the paper:
>
>
> >Contribution 2:  We discover that spurious correlations are formed primarily due to a handful of all the samples containing spurious features **through extensive empirical investigation**. Based on this insight, we propose a simple and novel data pruning technique that identifies and prunes a small subset of the data that contains these samples.
>
>
> Changes in contributions in Global Response to the Review Team:
>
> >Contribution 2: Discovering that spurious correlations are primarily formed from a handful of all samples containing the spurious features **through extensive empirical investigation**.
>
> >Contribution 3: Proposing a novel data pruning solution that severs spurious correlations in the **proposed** novel settings while attaining state-of-the-art performances on previously studied settings.
>
> \
> \
> Ye et. al. 2023 AISTATS “Freeze then Train: Towards Provable Representation Learning under Spurious Correlations and Feature Noise”, AISTATS, 2023.
>
> \
> We are happy to address any further concerns the reviewer may have. Thank you for your time and effort in improving our paper and thank you for your enthusiasm towards our paper.

---

> > ### Comment · Reviewer_X6Fn · 2024-11-25
> >
> > **Definintions**
> > I appreciate the authors updated definitions. The points I attempted to express were mixed together, and I incorrectly cited Liu et. al. 2021 as support.  I was trying to explain that it seems to me that *all* core features may not be required to be present in *all* samples of a given class---e.g., there could be particular traits that birds of class1 can have that no birds of class2 can have---and *some* background features could be present in *all* samples all samples---e.g., for a given sample, perhaps all the birds of class 1 just so happen to have the same background. This, though, is a philosophical argument about what it means truly be a core and background feature, so I think it makes sense for the authors to adopt their particular definition, and given it's reasonable (including the fact it is consistent with others in the literature) we operate from there.
> >
> > **Contexts Covered and Claims**
> > I appreciate the authors softening their claims, and I want to be clear that I completely agree with the author's response about the novelty and value of their work despite not providing theory, and that empirical work can/does spur new theory. To make the point salient I feel confident there exist theoretical and methodological tools that can be extended to formalize the solution authors have discovered, if it is in fact generally true, given the relationship of their result on data pruning to those in other settings I am very familiar with. The non-trivial work I still see is the proof of the generality of their observations, and under what conditions.
> >
> > I know the authors look at 5 datasets, and there are many things these datasets have in common that reasonably may mediate (or perhaps even define) the presence of their insights; I gave two as examples in my previous response: these are all image datasets (note this also separately constrains the architectures) and they seem to be easy prediction challenges. However, I was unaware the work covered text as well, I did a quick review of the paper again and unless I am missing the reference, they do not seem to make this clear. If I did not miss the reference, I would then recommend explicitly making this clear. I imagine other readers may not recognize this given the motivation and examples are all image-related. I am unsure if the authors already have resulted in textual data sets or if doing so would be simple. But if the authors can demonstrate the same set of discoveries with textual datasets as well, that would go a **long** way in making a case for the generality of the result.  As a result, I would be very happy to raise my score (considerably), especially in light of how negative some of the other reviews appear to be, for reasons I don't fully understand.

---

> > > ### Author Response · Authors · 2024-11-25
> > > **Response to Reviewer X6Fn (Part 4)**
> > >
> > > We thank the reviewer for their acknowledgement and continued efforts to improve our paper.
> > >
> > > Of the 4 non-synthetic tasks (non-testbed), one of them is a Language task, MultiNLI (Lines 244 - 249). This task further reinforces the generalizability of our claims and observations, as we use Transformer-based models (BERT) in this setting, which are significantly different from standard feed forward networks used in certain experimental settings in our paper. We make this clearer in the introduction of Section 4, where we emphasize that our study contains both Vision and Language Tasks.
> > >
> > > Kindly let us know if there are any remaining concerns and we will be happy to address them. Thank you.

---

> > > > ### Comment · Reviewer_X6Fn · 2024-11-26
> > > >
> > > > I thank the authors for their continued willingness to clarify their work. Given their use of text data to demonstrate generalizability, my final request to the authors for this paper in that pursuit would be to produce examples of Figures 4 and 5b using the MultiNLI data to show that the observations/insights in them are consistent within textual data. Separately, I'd also ask the authors to add text to the paper to make clear the difference between Figures 4 and 6, they appear to be very similar, so if they are attempting to convey different information, the nuance isn't clear to me.
> > > >
> > > > Relatedly, the authors say "In Fig. 4, we show that by simply excluding a few samples containing the spurious feature with hard invariant features in the CelebA setting studied in Sec. 3 (5% of all samples with spurious features in that class, 1% of the total train set), we observe significant improvements in Worst Group Accuracy" but later also say "we note that on pruning these samples, we do not observe significant drops in overall testing accuracies, implying that these samples do not contribute significantly to generalizability either (Fig. 4)." So they appear to be claiming Fig 4, which best I can tell can only show empirical evidence of pruning's joint impact on train and test accuracy from one dataset, is used as support for the insights about two datasets, one of which is "unidentifiable" while the other is "identifiable".
> > > >
> > > > **Future Suggestions**
> > > >
> > > > In the spirit of offering suggestions in general, below I lay out why I think the authors actually could go further than my requests above, and empirically demonstrate that **all** their major insights also exist in the textual data. Doing so, as I said prior, would go a very long way in demonstrating the generalizability of these insights. So if not for this work, perhaps something for them to consider in the future.
> > > >
> > > > As I understand it the authors make a set of inductive observations based on analyzing various datasets. To be clear, I do not believe the authors show every insight on every data set they directly could, but since 4 out of 5 are images perhaps they feel the results are somewhat transferable. Given they only have one text data set, it could be valuable to put some additional energy into finding/generating/augmenting text data that would allow them to show each of these insights.
> > > >
> > > >
> > > >
> > > > 1. Samples with simple core features do not contribute to spurious correlations.
> > > >
> > > > 2. Samples with hard core features are primary contributors to spurious correlations.
> > > >
> > > > To recreate 1 or 2 (Figure 2) with text data, the authors would need a "test bed" dataset that "does not contain significant spurious cues that can impact the difficulty of learning." I assume such a text dataset would exist that they could recreate their analysis by simulating easy to difficult spurious correlation by adding text appropriately (analogous to the addition of a line to the images in CIFAR-10). Perhaps they could even use the MultiNLI directly, and simply remove (or add) negation words in all its text samples to remove the spurious correlation, since "in the MultiNLI setting, the spurious feature comprises the same set of
> > > > negation words across the training data." If the decision was to remove negation words in the original MultiNLI, maybe simulating increasingly difficult spurious features could be reintroducing negation words at different frequencies (i.e., the number of negation words) in each sentence, or perhaps some other better way.
> > > >
> > > >
> > > >  3. Excluding a few key samples during training severs spurious correlations.
> > > >
> > > >
> > > > To demonstrate point 3 (Figure 4) one should be able to simply use the MultiNLI data directly. In fact, the authors claim that the results of Figure 4 are evidence for both the Celeb Data and also the MultiNLI. But I wonder if this was a mistake as Figure 6 seems to be a very similar graph also described as representing the Celeb Data?
> > > >
> > > >
> > > >  4. Spurious feature strength creates a specific distribution of the training data
> > > >
> > > >
> > > > To demonstrate point 4, Figure 5a could again be done with whatever text data is used in points 1 and 2.  Figure 5b could directly use the MultiNLI, as it is already an example of an identifiable case. I think the Multi can be adjusted to represent an unidentifiable case using the negation removal suggestions on MultiNLI mentioned for points 1 and 2. Specifically, I think one can make the MultiNLI have the same properties of the Celeb Data: one class can have all the examples of the spurious feature (men-glasses or contradicts-negation) and that proportion of class examples with that feature can be controlled (by how many contradicts examples the negation is moved from).
> > > >
> > > >
> > > >  5. Spurious Information is often unattainable
> > > >
> > > >
> > > > To demonstrate point 5, one can again make the MultiNLI have the same properties of the Celeb Data as described in point 4, and the directly produce the graph from Figure 2.

---

> ### Author Response · Authors · 2024-11-28
> **Response to Reviewer X6Fn (Part 5)**
>
> We thank the reviewer for their comments.
>
> **Clarifying Figures 4 and 6:**
>
> In Fig. 4, we only prune those samples that contain spurious features and hard core features. This is to make the point that samples with hard core features + spurious features are the primary contributors to spurious feature reliance. Through Fig 6, we propose the final data pruning algorithm which prunes all the samples with hard core features (which contains samples with hard core features + spurious features and also sample with hard core features + no spurious features, Lines 435 - 438), as spurious information is unidentifiable. We have added additional text to make this clearer on Line 436.
>
> ---
> \
> **Presenting the same insights with textual data:**
>
> **Insights 1 and 2:**
>
> **Samples with simple core features do not contribute to spurious correlations. Samples with hard core features are primary contributors to spurious correlations.**
>
> To show this, we perform the same experiment in Section 4, but instead of CIFAR-10S, we use the MultiNLI dataset. First, we remove all samples with negation words from the **training** data and then we compute the sample-wise difficulty scores as we do for CIFAR-10S in Section 4. We then create two settings: one where we introduce the spurious negation word “never” at the end of the 100 hardest input samples belonging to class 1 (**contradicts**) and another where we introduce the spurious negation word “never” at the end of the 100 easiest input samples belonging to class 1 (**contradicts**). We do the same to a set of test samples belonging to class 2 (**neutral with**) and class 3 (**entailed by**).
>
> Consistent with the standard MultiNLI setting, we measure the degree of spurious feature reliance through Worst Group Accuracy (accuracy of the set of test samples of class 2 or class 3 with the spurious feature).
>
> We observe that WGA is **significantly** worse when the word “never” occurs in the hardest samples vs. the easiest samples during training.
>
> Introducing spurious feature in easiest 100 samples: WGA = **55.22%**
>
> Introducing spurious feature in hardest 100 samples: WGA = **1.04%**
>
> A higher WGA indicates low reliance on spurious features.
>
> The gap in worst group accuracy is 54.18%. Note that the number of samples containing the spurious feature is the same in both settings (= 100).
>
> Additionally, we note that there are 191,504 training samples in this setting. There are 57,498 samples belonging to the **contradicts** class. We introduce the spurious feature in only 100 samples of the **contradicts** class (0.17% of samples within the class, 0.0522% of all samples in the training set.) We also observe that in a setting with no spurious features during training, Worst Group Accuracy is 67.42%.
>
> Simply varying which 100 samples contain the spurious negation word “never” has such a **huge** impact on Worst Group Accuracy. This finding is extremely insightful, novel and is consistent with the results observed in Section 4 (Figure 2) of our paper.
>
> This experiment reinforces the claim that samples with hard core features are primary contributors to spurious correlations and that samples with simple core features do not contribute to spurious correlations.
>
> ---
> \
> **Insight 3:**
>
> **Excluding a few key samples during training severs spurious correlations.**
>
> For this, we simply show the results in Figure 4 but for the MultiNLI dataset. Note: Due to computational constraints, we only show three pruning sparsities.
>
> Worst Group Accuracy:
>
> | Prune %   | 20%| 25%| 33.33%|
> | -------- | ------- | ------- | ------- |
> |Pruning Easiest| 66.81| 65.59| 65.33|
> |Pruning Hardest| 72.21| 73.17| 76.05|
>
> Kindly note that the model attains 65.9% Worst Group Accuracy on the original, unpruned dataset.
>
> We refer the Reviewer to Fig. 10 in the appendix for a better understanding.

---

> ### Author Response · Authors · 2024-11-28
> **Response to Reviewer X6Fn (Part 6)**
>
> **Insight 4:**
>
> **Spurious feature strength created a specific distribution of the training data.**
>
> To show this, we use a smaller subset of the MultiNLI dataset and vary the strength of the spurious signal by varying the proportion of samples containing the spurious feature.
>
> Distribution of samples with spurious features in Identifiable setting:
>
> | Q1 (Easiest) | Q2 | Q3 | Q4 (Hardest) |
> | ------- | ------- | ------- | ------- |
> |57.4% | 24.3% | 11.5% | 6.8% |
>
> Distribution of samples with spurious features in Unidentifiable setting:
>
> | Q1 (Easiest) | Q2 | Q3 | Q4 (Hardest) |
> | ------- | ------- | ------- | ------- |
> |28% | 21% | 24% | 27% |
>
> To confirm that the unidentifiable setting still causes the network to rely on spurious correlations, we create another setting where we remove all samples with spurious features and compare the Worst Group Accuracies below:
>
> Unidentifiable setting: 64.89%
>
> No samples with spurious features setting: 70.73%
>
> We observe that in the setting with no samples containing the spurious features, Worst Group Accuracy is higher, indicating that the unidentifiable setting still causes the network to rely on spurious correlations but the samples containing spurious features are uniformly distributed.
>
> ---
> \
> **Additional Results:**
>
> **Distribution of the MultiNLI setting covered in the paper (Identifiable) (Figure 5(b) extension).**
>
> Distribution of samples with spurious features:
>
>
> | Q1 (Easiest) | Q2 | Q3 | Q4 (Hardest) |
> | ------- | ------- | ------- | ------- |
> |44.39% | 23.83% | 16.68% | 15.08% |
>
> Q1 in this setting contains almost half of all samples with spurious features while Q4 only contains 15%. This shows that samples with spurious features are not uniformly distributed when viewed through the lens of sample difficulty. This is in contrast to the CelebA setting (Unidentifiable), in which samples with spurious features are uniformly distributed (Figure 5(b) right in text).
>
> We refer the Reviewer to Fig. 11 in the appendix for a better understanding.
>
> ---
> \
> Kindly let us know if there are any additional concerns that we can address which can help increase our score. We greatly appreciate your detailed comments that have helped improve our paper. Thank you again for taking the time to review our work.

---

> > ### Comment · Reviewer_X6Fn · 2024-11-29
> > **Thank You**
> >
> > I thank the authors for their continued willingness and thoroughness to address the questions I've raised. I hope they feel like it has improved their work. While the most important discoveries do seem to replicate, the fact that not all do, I think gives the literature (or the authors themselves) additional fodder for digging deeper and understanding what is truly generally true, or at least under what conditions. I myself have curiosities (I feel deeply invested in understanding this question now) but I think the results from this additional analysis with the text data convince me (as much as is possible from purely empirical experiments) that the authors have truly discovered some valuable generalizable truths. I again thank the authors, congratulate them on a truly awesome paper, and will raise my score accordingly!

---

> > > ### Author Response · Authors · 2024-11-29
> > >
> > > We thank the reviewer for their tireless effort towards our work! We sincerely thank them for taking the time to understand our work in depth and present detailed comments and suggestions that have definitely helped improve our paper. We are grateful that the reviewer recognizes the key insights in the paper that we are excited about. Thank you again for your time and effort.

---

### Author Response · Authors · 2024-11-18
**Global Response**

We thank the reviewers for their comments. We are glad that the reviewers recognize the key contributions of the paper:

**Contributions:**
1) Identifying and targeting novel settings where obtaining spurious information is difficult/impossible and showing the failure of past techniques on these settings.
2) Discovering that spurious correlations are primarily formed from a handful of all samples containing spurious features through extensive empirical investigation.
3) Proposing a novel data pruning solution that severs spurious correlations in the proposed novel settings while attaining state-of-the-art performances on previously studied settings.

We believe we have addressed the concerns of all reviewers by responding to them individually. We summarize the resolution of the primary concerns that were shared by the reviewers:

**Weaknesses Resolved:**

1) Included more results across more architectures and hyperparameters to further reinforce our claims and observations. (Reviewers eCbW, X6Fn)
2) Improved all relevant formal definitions/more detailed figure captions/technical details. (Reviewers 7BQb, n9k8)
3) Released our code through an anonymized github link and will open-source it upon acceptance. (Reviewer eCbW)


However, we strongly believe that the score received from reviewer 7BQb is not a fair assessment of our work. Their only concern was the clarity or presentation of the paper but they have given our contribution and soundness a low score as well. We would like to emphasize that the clarity and writing quality of the paper were well received by all other reviewers. Nevertheless, we have addressed the clarity concerns of reviewer 7BQb and hope that our work is now well received. If not, we are happy to address any further concerns.

---

### Comment · Area_Chair_N8q2 · 2024-11-19
**Discussion Phase**

Dear Reviewers,

Please review the authors' replies and consider the feedback from other reviewers. If your concerns remain unresolved, feel free to ask for further clarifications. We have until November 26th for discussion.

Thank you for your efforts.

Best regards,
Area Chair

---

> ### Comment · Area_Chair_N8q2 · 2024-11-23
>
> Dear Reviewers,
>
> As we near the end of the discussion period, this is your last chance to engage.
>
> Please review the authors' replies and feedback from other reviewers. If any concerns remain, request clarifications. The authors have one more day to respond before the reviewer-only discussion week begins.
>
> Thank you for your efforts.
>
> Best regards,
> Area Chair

---

### Author Response · Authors · 2024-12-03
**Global Response for the Area Chair and the Reviewers**

We thank the reviewers for their comments, which have helped improve the quality of this paper. We strongly believe that we have addressed all concerns raised by all reviewers. Below, we summarize the discussion at a high level:

---

>**Summary**

The reviewers recognized that this work tackles an important and critical problem (X6Fn, 7BQb) through a very intuitive solution (X6Fn), contains novel contributions (n9k8, X6Fn), delivers interesting and exciting insights and observations (X6Fn), discovers some valuable generalizable truths (X6Fn), contains simple and compelling experiments (X6Fn), provides a thorough experimental design with multiple datasets (eCbW), and that it was well written (n9k8, X6Fn) and well organized (eCbW, n9k8, 7BQb). While some reviewers had some initial comments regarding clarity of the paper (7BQb, n9k8), these have now been addressed and acknowledged by the reviewers.

---

The following are comments that we have addressed but have **not** heard back regarding:

1) Reviewer **eCbW**: Reviewer eCbW asked us to **open source our code, which we have**. They point out that the settings studied in this paper are synthetic and that we should include settings with real-world spurious features. **This is incorrect** as only 1 out of the 5 datasets studied has synthetic spurious features which is **necessary and intentional** as it enables us to alter the synthetic spurious feature to draw insights. The other four datasets contain **real-world** spurious features in real-world settings. Furthermore, apart from the novel settings proposed, the standard settings studied are consistent with those in the literature. We have also received **no references or settings** that they want us to reproduce our results in. They also asked us to provide **additional results across different architectures and hyperparameters, which we have**. However, we have not heard back from them at all during the rebuttal phase.

2) Reviewer **7BQb**: As we have stated in our last response, we believe Reviewer 7BQb may have misunderstood our explanation on what impacts the strength of a feature. There are many factors that can be varied to alter the strength of a feature. In our paper, the strength is **varied** by two factors: frequency and area. They additionally claim that these concepts are not well defined. We emphasize, however, that these concepts are heavily studied in all relevant literature that we have cited.
We also note that discovering and considering all factors that can impact the strength of a feature is beyond the scope of this paper and that this does not impact the generalizability of our findings, as they ultimately alter the same thing: the strength of a feature. We have already shown that frequency and area can be modified interchangeably to alter the strength. One could also modify the strength by making red color in eyeglasses a lighter/darker shade, as the reviewer has pointed out. For instance, one could vary the shade to create the same plots in Figure 5 (a) but this will **not offer any new insights**. It is also unclear why the reviewer believes that their example of eyeglasses impacts generalizability and relates to Reviewer X6Fn’s comments. However, we have not heard back from them.

---

### Meta-Review · Area_Chair_N8q2 · 2024-12-19

**Metareview:**

### Summary

The authors investigate spurious correlations in a controlled environment by introducing them into existing datasets. Through a series of calibrated experiments, they identify how individual samples contribute to spurious correlations. This leads to a pruning strategy that significantly improves worst-group performance without drastically reducing average performance. The proposed method achieves state-of-the-art results compared to other strategies.

### Strenghts

The paper is well-written and easy to follow.
The authors guide the reader through their experiments, elucidating their findings and the reasoning behind the method.
This paper gives both a contribution to the understanding of the phenomenon and to the development of a mitigation strategy.

### Weaknesses

No major concerns.
Marginal improvements over other mitigation strategies.

### Reasons for acceptance

This paper presents a simple yet impactful idea, supported by experiments, that leads to a practical algorithm. All major critiques have been addressed, and I see no reason for rejection.

**Additional Comments On Reviewer Discussion:**

* **Clarity:** Reviewers raised several questions about the core ideas and interpretation of the results. The authors provided convincing responses, conducted additional experiments, and updated their submission accordingly.
* **Limited experiments:** Though limited in benchmarks and architectures, the experiments align with related works. The authors also expanded their results.
* **Reproducibility:** Initially criticized for not sharing code, the authors have since released it.

The discussion was extensive, and some reviewers stopped engaging, but I believe their concerns were addressed in the final replies.

---

### Decision · Program_Chairs · 2025-01-22

Accept (Spotlight)